# Mid-cell migration of the chromosomal terminus is coupled to origin segregation in *Escherichia coli*

Ismath Sadhir[1,2] & Seán M. Murray [1] ✉

Bacterial chromosomes are dynamically and spatially organised within cells. In slow-growing *Escherichia coli*, the chromosomal terminus is initially located at the new pole and must therefore migrate to midcell during replication to reproduce the same pattern in the daughter cells. Here, we use high-throughput time-lapse microscopy to quantify this transition, its timing and its relationship to chromosome segregation. We find that terminus centralisation is a rapid discrete event that occurs ~25 min after initial separation of duplicated origins and ~50 min before the onset of bulk nucleoid segregation but with substantial variation between cells. Despite this variation, its movement is tightly coincident with the completion of origin segregation, even in the absence of its linkage to the divisome, suggesting a coupling between these two events. Indeed, we find that terminus centralisation does not occur if origin segregation away from mid-cell is disrupted, which results in daughter cells having an inverted chromosome organisation. Overall, our study quantifies the choreography of origin-terminus positioning and identifies an unexplored connection between these loci, furthering our understanding of chromosome segregation in this bacterium.

The faithful and timely segregation of the replicated chromosomes is an essential step in every bacterial cell cycle. While in some bacteria this can be directly attributed to the well-studied ParABS partitioning system[1,2], in other species this system is either not strictly essential[2–4] or is absent altogether[5,6]. In particular, the mechanism of chromosome segregation in the model system *Escherichia coli* has yet to be identified.

Whatever the mechanism, segregation of the chromosome very likely goes hand in hand with its organisation. Indeed, rod-shaped bacteria have their chromosome arranged linearly within the cell such that the position of each chromosomal locus can be predicted from its genomic position[7–10]. However, the orientation within the cell can differ between species and conditions. Typified by *Caulobacter crescentus*, the longitudinal arrangement has the replication origin (*ori*) and terminus (*ter*) located at opposite ends of the cell with the two chromosomal arms organised linearly along the long cell axis[8,9,11,12]. On the other hand, in slow-growing *E. coli* cells, the unreplicated chromosome

is believed to adopt a transverse organisation in which the origin (*ori*) is positioned at mid-cell with the left and right chromosomal arms on either side and a stretched terminus region between them[13,14]. Upon replication, the duplicated *ori* segregate outward to the quarter positions with the other replicated loci following progressively, resulting in each replicated chromosome having the same left-*ori*-right organisation and therefore inheritance of the pattern by the daughter cells. However, the symmetry of the above pattern is initially broken by *ter*, which in newborn cells is located close to the new pole i.e. not close to *ori* as might be expected for a circular chromosome. Inheritance of the birth pattern is then achieved by the migration of *ter* to midcell (*ter* centralisation) during the cell cycle[15,16].

The *ter* sits within the 800 kb Ter macrodomain defined in part by the presence of 23 *matS* sites[17,18]. The protein MatP binds to these sites and displaces MukBEF, the *E. coli* Structural Maintenance of Chromosomes complex[19], resulting in a decrease of long-range

[1]Max Planck Institute for Terrestrial Microbiology and LOEWE Centre for Synthetic Microbiology (SYNMIKRO), Marburg, Germany. [2]Microcosm Earth Center, Max Planck Institute for Terrestrial Microbiology and Philipps-Universität Marburg, Marburg, Germany. ✉e-mail: sean.murray@synmikro.mpi-marburg.mpg.de

chromosomal contacts within the region, consistent with its less condensed organisation[20]. MatP has been shown to bridge DNA at *matS* sites[17]. However this is easily out-competed by non-specific DNA binding and so may not be relevant in vivo[20,21].

As described above, *ter* is initially located near the new pole but moves to mid-cell during chromosome replication. Its maintenance there is believed to be due to a protein linkage connecting MatP to the divisome protein FtsZ[22,23] which facilitates the resolution of chromosome dimers by the FtsK/XerCD machinery just before cell division[21,24]. It may also allow *ter* to act as a positive regulator of divisome positioning[25]. Despite these studies, the cause of *ter* centralisation and its role within the segregation process has remained unclear.

Here, we use high-throughput single-cell imaging and analysis to quantitatively establish the choreography and timing of *ori* and *ter* in slow-growing *E. coli* cells. We find that *ter* migration from the new pole to mid-cell occurs ~25 min after initial separation of replicated *ori* and independently of its linkage by MatP to the divisome (the Ter-linkage). The movement is rapid, occurring within ~5 min, and is coincident with the arrival of *ori* to the quarter-cell positions suggesting a coupling between these two events. Consistent with this, *ter* is unable to stably localise at mid-cell in cells with impaired origin segregation. Overall then, our results show that the stable mid-cell positioning of the terminus region is due to a previously unknown coupling to origin segregation.

## Results

### The cell cycle dynamics of *ori* and *ter*

The accurate analysis of *ori* and *ter* dynamics requires the temporal imaging of a large number of cell cycles. We achieved this using a high-throughput single-cell approach based on a 'mother-machine' microfluidic device[26,27]. Together with a custom analysis pipeline[26], this allowed us to segment and track tens of thousands of cell cycles in steady-state conditions (Fig. 1a, Supplementary Fig. 1a). To visualise the *ori* region, we used the P1 ParABS labelling system (consisting of $parS_{P1}$ inserted near *oriC* and an mTurquoise2-ParB$_{P1}$ fusion expressed from a plasmid[28]). We used this monomeric fluorophore after we found that CFP-ParB[13] produces artefacts (see below). The terminus was visualised using a MatP-YPet fusion expressed from its endogenous locus, which has been shown to colocalise well with markers of the terminus region[18,22,25,29]. Cells grew in the device with a mean cell cycle duration of 133 min and mean birth length of 1.71 μm (Supplementary Fig. 1b, e) and under these conditions we were able to image the cells every 5 min while maintaining sufficient signal and without significant changes in growth rate (Supplementary Fig. 1d).

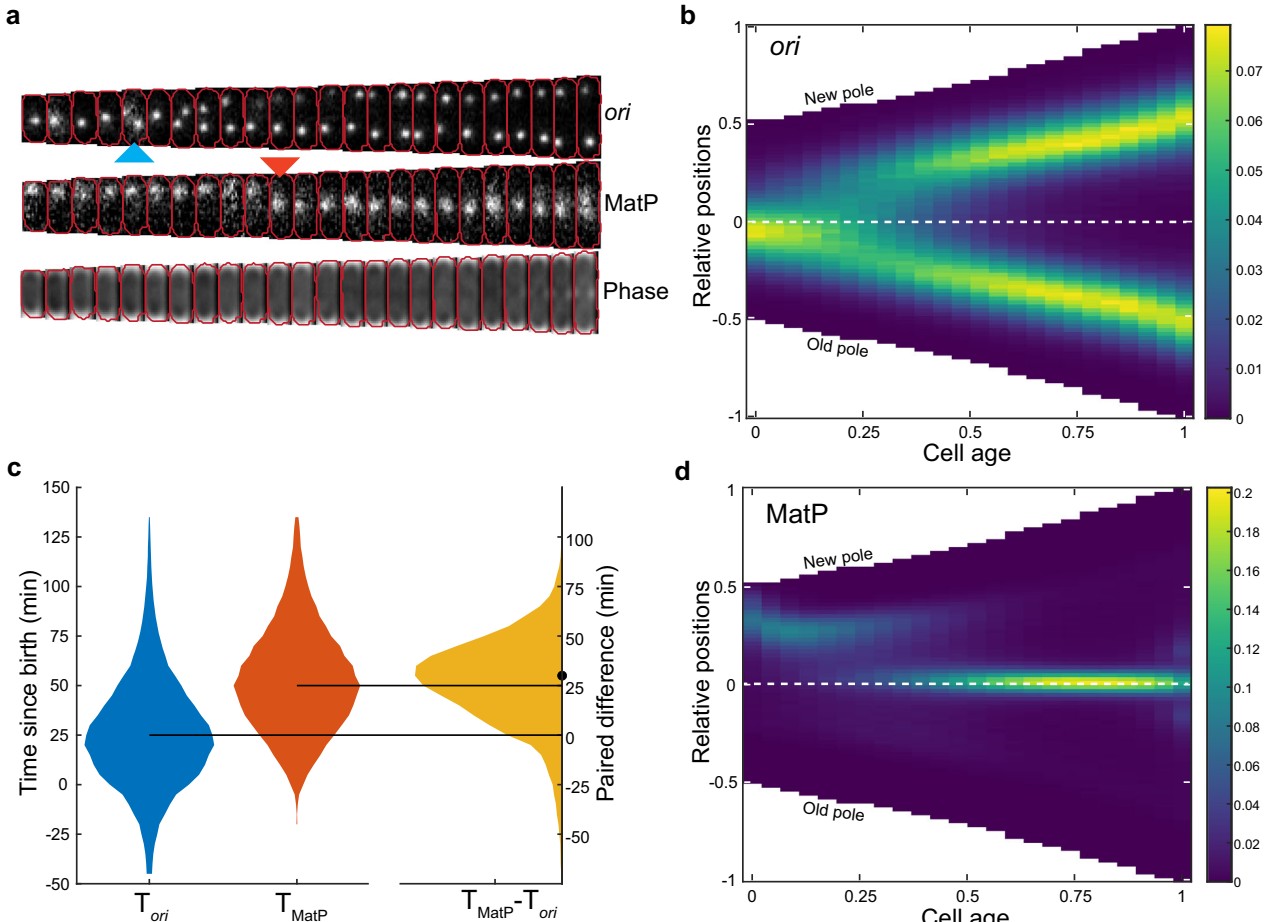

**Fig. 1 | MatP-labelled *ter* re-localises to mid-cell after *ori* focus duplication. a** An example cell cycle with *ori* (mTurquoise2-ParB$_{P1}$) duplication and *ter* (MatP-YPet) relocalization frames marked with blue and red arrows respectively **b** Population kymograph of *ori* foci positions along the long axis of the cell. Data from different cell cycles are combined by using cell age (0 is birth, 1 is division) and positions relative to an exponentially increasing normalised length. **c** Distribution of the time of *ori* focus duplication (mean ± sd = 26.6 ± 28.0 min) and MatP relocalization (52.8 ± 26.1 min) along with the time difference between the two events

(26.5 ± 23.2 min). The negative values in $T_{ori}$ and $T_{MatP}$ correspond to events occurring in the previous cell cycle. The horizontal lines indicate the median values of 25 and 50 min for $T_{ori}$ and $T_{MatP}$, respectively. The dot indicates the median (30 min) of the paired difference $T_{MatP}-T_{ori}$. **d** Population kymograph of foci positions of MatP positions as in **b**. The values in the colour scale for the kymographs represent the frequency of occurrence of foci positions normalised to the number of cell cycles at each cell age. *n* = 38066 cell cycles. Source data are provided as a source data file.

We first confirmed the cell cycle dynamics of *ori*. Consistent with previous studies using agarose pads at similar growth rates[15,16,28,30], newborn cells typically had a single *ori* focus close to midcell that, upon duplication, segregated outwards to the quarter positions (Fig. 1b). The high-throughput and temporal nature of our data allows us to quantify these observations in detail. Separation of *ori* (defined by two *ori* foci seen for the first time in the cell cycle) occurred on average 27 min after birth, at a cell length of 2.0 μm but with substantial variation between cells (Fig. 1c, Supplementary Fig. 1l). Indeed, we found that 15% of cells were born with more than one *ori* focus (Supplementary Fig. 1f), indicating that DNA replication was initiated in the previous cell cycle (note that this is not detectable in the population average kymographs (Fig. 1b) or demographs (Supplementary Fig. 1g)). We have shown elsewhere that this is consistent with the volume dependence of chromosome replication initiation[31,32] and arises from a second replication initiation in the mother cell due to the size of the mother cell crossing the volume per origin initiation threshold for a second time in the same cell cycle[33].

We also confirmed the positioning of *ter* - it was found close to the new pole at birth before moving to midcell, where it remained tightly localised for the majority of the cell cycle (Fig. 1d). To quantify the timing of this transition, we defined *ter* arrival at midcell as the first frame at which the MatP-YPet focus is within the middle 4.8 pixels (320 nm) of the cell for 3 consecutive frames. The former value was based on the position distribution (Supplementary Fig. 1k), while the latter was an arbitrary choice that sought to strike a balance between avoiding transient centralisation events and being insensitive to missed foci detections. The same values are used throughout this study. Using this measure, stable *ter* localisation to mid-cell occurred on average 53 min into the cell cycle and 26 min after *ori* separation (Fig. 1c). These timings are consistent with those previously inferred from snapshots[30]. Here, however, we follow complete cell cycles and have captured the entire distribution of timings.

The visible separation of *ter* foci occurred just before cell division consistent with its association to the early divisome protein FtsZ, though the timing of this relative to the cell cycle is dependent on our identification of the cell division event. Correspondingly, at birth *ter* foci were initially found closer to the pole before moving inward to the edge of the nucleoid (Fig. 1a, d and below).

## Origins and nucleoid are asymmetrically positioned

Our data also show that *ori* is not precisely positioned at the mid- and quarter-cell positions but rather exhibits a bias towards the old pole at birth and to either pole before division. The offset is small (about 5% of cell length) but reproducible and persistent during the beginning of the cell cycle. As a consequence, the trajectories of segregating *ori* foci are not symmetric with the new-pole proximal *ori* moving further and faster (see below) to reach its target quarter cell position. Note that the bias at birth is only apparent when cells are ordered according to their polarity. It is not detectable when cells are oriented randomly, as may be the case for a snapshot-based analysis (Supplementary Fig. 2a). The mid-cell positioning of *ter*, on the other hand, is precise (Fig. 1d).

Since the nucleoid exhibits a new-pole bias during the early part of the cell cycle[34–36], we sought to determine how *ori* is positioned relative to the nucleoid. We therefore examined the localisation of *ori* and *ter* in a strain expressing the nucleoid marker HU-mCherry (Fig. 2a). Consistent with previous results, we found a clear bias of the nucleoid toward the new pole that gradually decreases during the first half of the cell cycle until the nucleoid is symmetrically positioned within the cell (Fig. 2b). As a result, at birth the *ori* is positioned at the old-pole proximal periphery of the nucleoid, typically at the outer quarter mark of the (background-subtracted) HU-mCherry signal. After duplication, one *ori* moves to the opposite side of the nucleoid resulting in a symmetric configuration both with respect to the nucleoid and the cell (Fig. 2c). Interestingly, the position of *ter* is unaffected by the initial

bias in the nucleoid position, perhaps because the bias has largely been resolved by the time of *ter* centralisation (Fig. 2d).

Overall these results refine our understanding of *ori* and *ter* positioning in slow-growing cells. In terms of positioning within the cell, our analysis reproduces the established view, largely based on snapshot imaging[15,16,28,30,37], that *ori* exhibits mid-cell positioning at birth, albeit with the addition of a slight old-pole bias, followed by segregation to the quarter positions. However, it has been assumed that nucleoid is centrally positioned so that the *ori* lies in the middle of the nucleoid (or nucleoid lobe), consistent with an observed left-right chromosome organisation[13,14]. This is not what we observe in our conditions - the *ori* lies toward the nucleoid periphery both before initiation at the beginning of the cell cycle and after segregation. This agrees with earlier studies using FISH and DAPI-labelling[34,35,38,39] that found that the chromosome is organised longitudinally i.e. *ori* and *ter* at opposite ends of the nucleoid with the two chromosomal arms between them.

To investigate this further, we used another *parS*/ParB labelling system, pMT1[13], to separately label each of the two chromosomal arms close to the genes *elaD* and *rhlE* and at approximately 240° and 120° respectively. These loci are close to regions (L3 and R3) previously identified as being spatially distant[14]. The *ori* and nucleoid were labelled as above. Upon imaging these strains, we found similar positioning of the *ori* relative to the nucleoid i.e. close to the old-pole-proximal/outer periphery of the nucleoid (Supplementary Fig. 3a, b, d, e). For unknown reasons, the right-arm labelled strain displayed earlier *ori* duplication but this did not affect its overall organisation. In contrast, both left and right loci were found toward the opposite edge of the nucleoid or nucleoid lobes (Supplementary Fig. 3c, f). This was clearest before division when the loci were typically detected at the opposite (inner) quarter mark of the corresponding nucleoids. Interestingly after duplication both sister loci maintained, on average, the same absolute position within the cell, unlike *ori*, which maintains the same relative position (at the cell quarters). Similar results were obtained when we gave up labelling *ori* to simultaneously label the left and right loci with *parS*$_{P1}$ and *parS*$_{pMT1}$ respectively (Supplementary Fig. 3g–i).

To quantify the observed organisation, we measured the frequency of the different possible arrangements of *ori* (O), *elaD* (L) and *rhlE* (R) relative to the new (NP) and old (OP) poles. We found that 84% or 83% of new-born cells showed an NP-L-O-OP or NP-R-O-OP pattern respectively i.e. the left and right loci are predominantly found closer to the new pole than *ori* (Supplementary Fig. 3j, k). Consistent with this, before division 78% or 74% displayed an O-L-L-O or O-R-R-O pattern respectively, thereby reproducing the predominant pattern in the daughter cells. We also measured the distance between the loci in the three strains and found that L and R are on average more separated from each other than from *ori* (Supplementary Fig. 3m, n). These findings are not consistent with a transverse L-O-R model of the chromosome[13,14,40,41] in which L and R are equally likely to be new-pole proximal at birth and in which the sister chromosomes are primarily related by translation i.e. an L-O-R-L-O-R pattern before division (Supplementary Fig. 3l). Rather, the data are broadly consistent with a longitudinal-like organisation in which the *ori* and *ter* are located towards opposite ends of the nucleoid with the two arms between them. We use the term 'longitudinal-like' to contrast with bacterial species such as *Caulobacter crescentus*[42,43] and *Myxococcus xanthus*[44], in which the *ori* is physically anchored in place and resides at the true nucleoid edge rather than at the outer quarter-mass position as we observe. Overall, our results show that *E. coli* can, as already observed for fast growth[45,46], have a longitudinally organised chromosome also during slow growth. We emphasise that as we have no reason to question the previous studies supporting a transverse organisation, our interpretation is that both transverse and longitudinal organisations can occur during slow growth (indeed, we observe a more

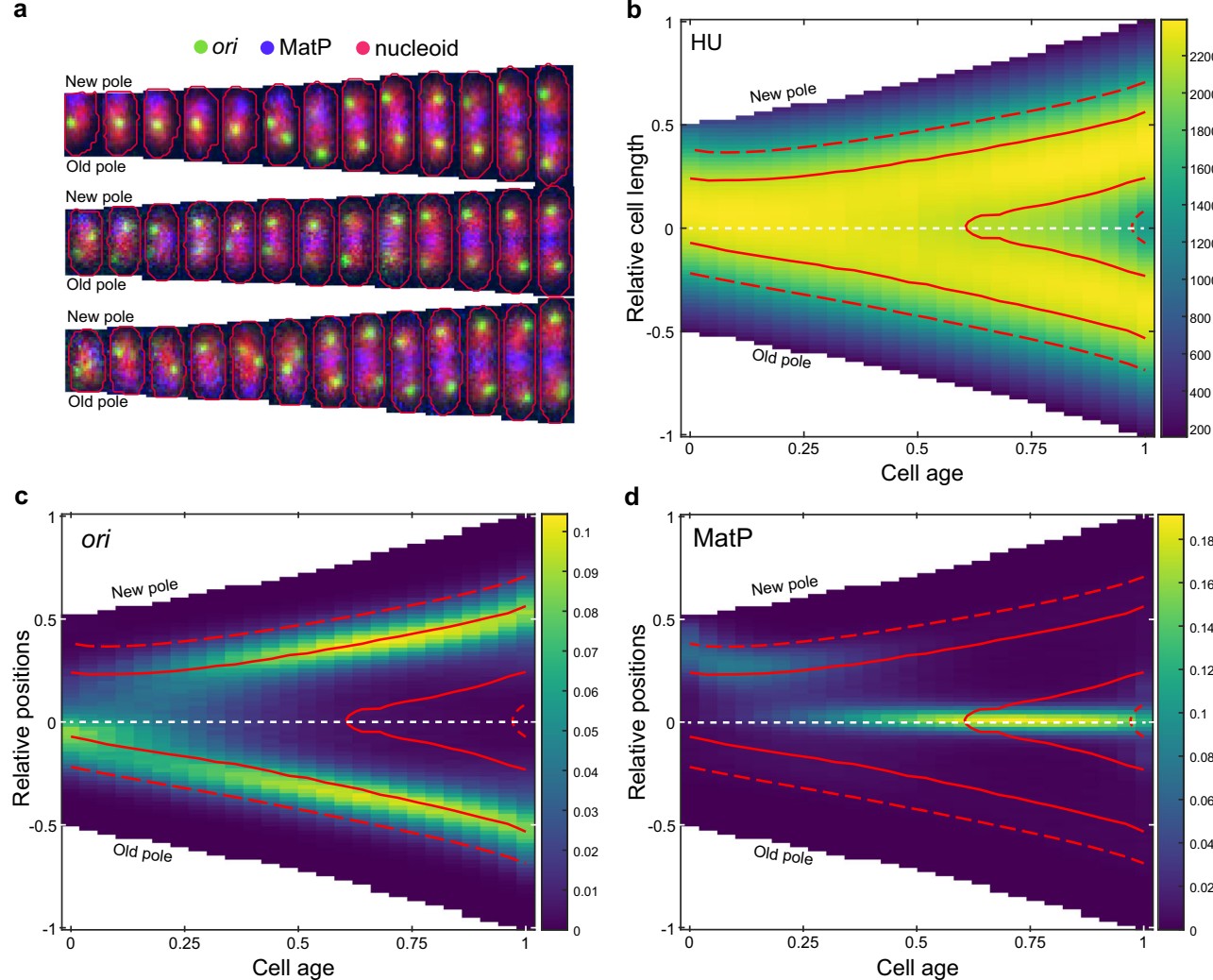

**Fig. 2 | The *ori* is positioned at the periphery of the nucleoid. a** Representative cell cycles (10-min intervals) with *ori* in green, HU in red and MatP in blue **b** Population kymograph of HU-mCherry signal along the long axis of the cell. The solid and dashed contour lines enclose the upper 50 and 80 percent respectively of the total HU-mCherry signal. **c** Population kymograph of *ori* foci positions (as in

Fig. 1b) with contour lines from **b**. **d** Population kymograph of MatP foci positions (as in Fig. 1d) with contour lines from (**b**). The values in the colormap scale for (**b**) represents the mean intensity of line profiles across the cell length. The colour scale in (**c**) and (**d**) are as in Fig. 1. *n* = 33593 cell cycles. See also Supplementary Fig. 2.

transverse organisation at the time of *ori* duplication (Supplementary Fig. 3j, k)). However, the conditions giving rise to one or the other pattern are unclear (see discussion below).

### *ter* centralisation occurs before nucleoid constriction

It was previously suggested that the stable appearance of a constricted/bilobed nucleoid structure occurs ~8 min after *ter* centralisation[29]. To examine this, we analysed the HU-mCherry signal across the long-axis of the cell and identified the time of stable (i.e. not transient) nucleoid constriction (Fig. 3a, Supplementary Fig. 4a, b). In agreement with Männik et al., we found that stable constriction occurs after *ter* centralisation. However we found a much longer interval (45 min) between the two events (Fig. 3b) suggesting no direct causal relationship between them. Indeed, it was recently shown that the relative timing of nucleoid constriction within the cell cycle depends on the growth medium[47]. Nevertheless, we have now identified the timing of three important cell cycle events—separation of duplicated origins, *ter* centralisation and the onset of nucleoid constriction (Supplementary Fig. 4c). While there is significant variation in the timing of these events between cells, in almost all cells they occur in the given order (Supplementary Fig. 4d).

### Stable *ter* centralisation coincides with completion of *ori* segregation

The kymographs and demographs of MatP positions (Fig. 1d, Supplementary Fig. 1h) reveal separated peaks, indicating that the migration of *ter* from the edge of the nucleoid to midcell is relatively rapid compared to its movement at other times in the cell cycle. In fact, this transition could occur within a single frame (5 min) (Fig. 1a, Supplementary Fig. 5a). This could be made apparent at the population level by synchronising the cell cycles according to the time of *ter* centralisation (Fig. 4a). Consistent with this, the mean step-wise velocity of MatP foci (measured between consecutive frames) sharply peaked at the transition before dropping rapidly to zero afterwards (Fig. 4b). Notably, this is not a consequence of the synchronisation, as the greatest movement most frequently occurred at the transition (Supplementary Fig. 5d).

While the migration of *ter* to mid-cell has previously been attributed to the action of the mid-cell localised replication machinery[22], it is unclear if this is consistent with such a rapid transition. Indeed a study of chromosome organisation during fast (1 h) growth found that *ter* centralisation was correlated with cell length rather than progression of the replication fork and that, irrespective of when the transition

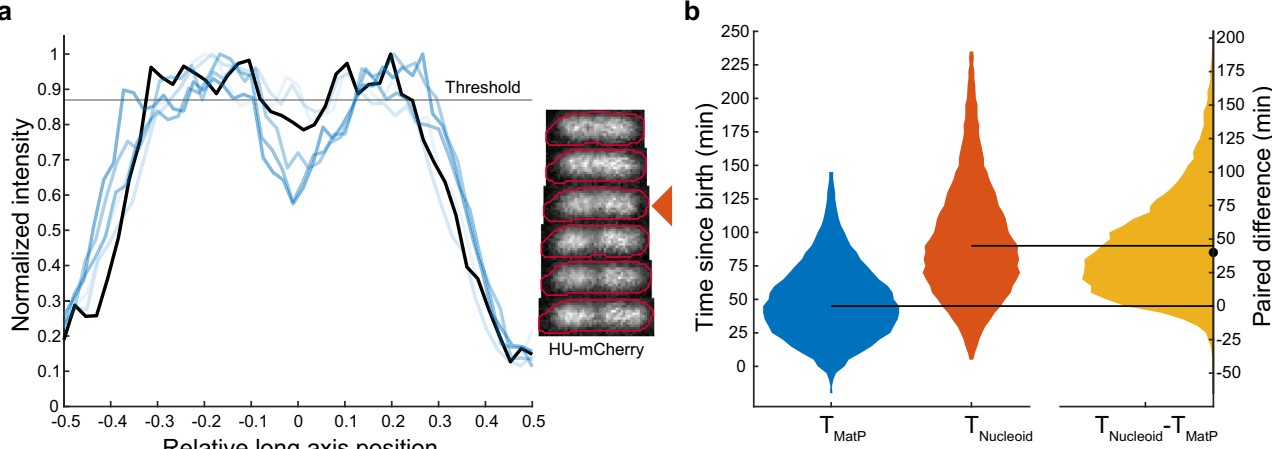

**Fig. 3 | MatP relocalization precedes nucleoid constriction. a** Line profile plots of HU-mCherry signal on different frames with corresponding images. Increasing darkness of blue indicates later times. The black line indicates the profile from the frame at which the nucleoid constricts. The nucleoid is considered constricted if the relative depth of the dip in the signal (if one exists) is greater than a threshold chosen to account for normal variation in the signal (see Supplementary Fig. 4a, b). The threshold for the frame indicated by the arrow is shown in red. The time of stable nucleoid constriction $T_{Nucleoid}$ is the earliest frame after which a dip greater than the threshold is observed in the nucleoid signal for the rest of the cell cycle. **b** Distribution of the time of MatP centralisation, $T_{MatP}$ (mean ± sd = 49.5 ± 28.4 min) and stable nucleoid constriction, $T_{Nucleoid}$ (94.2 ± 43.7 min) along with the time difference between the two events (51.5 ± 38.1 min) as in Fig. 1. The horizontal lines indicate the median values of 45 and 90 min for $T_{MatP}$ and $T_{Nucleoid}$ respectively. The dot indicates the median (40 min) of the paired difference $T_{Nucleoid}-T_{MatP}$. Data as in Fig. 2. See also Supplementary Figs. 2, 4. Source data are provided as a source data file.

occurred the remaining unreplicated DNA migrates with it[46]. This is consistent with the large variation (coefficient of variation (CV) of 0.88) we observe in the timing of the transition (Fig. 1C), which can occur even before visible origin separation or as late as 75 min afterwards. This is also much larger than the variation (CV of ~0.15) of the C-period (the time between replication initiation and termination) in comparable conditions[48]. Therefore, we find it unlikely that the timing of *ter* centralisation is set by replication fork progression.

On the other hand, the kymographs in Fig. 1 indicate that *ter* centralisation occurs at approximately the same time as *ori* segregation. In fact, after synchronising the *ori* foci positions relative to *ter* centralisation, it became clear that centralisation is coincident with the completion of *ori* segregation, i.e, with the arrival of the replicated *ori* at the quarter positions of the cell (Fig. 4c). The average velocity of both the new pole and old-proximal *ori* increased steadily up to the *ter* transition before dropping rapidly, with the peak occurring at the same time as that of *ter* (Fig. 4d). We additionally note that the new pole-proximal *ori* exhibits a higher mean velocity than its sister, consistent with our previous observation of asymmetric *ori* segregation (Fig. 2). Overall these results indicate a coupling between *ter* centralisation and the completion of origin segregation and it is tempting to speculate a causative relationship between the two events, namely that the final stage of *ori* segregation somehow triggers *ter* to rapidly move to midcell. To be clear the alternative hypothesis that *ter* centralisation enforces or signals the completion of *ori* segregation is also possible.

### *ter-ori* coupling does not depend on the *ter* linkage

If *ter* centralisation and *ori* segregation are genuinely coupled, then disrupting one or the other process may provide insight into their codependency. In this direction, we first targeted *ter* centralisation. As discussed above, while the cause of *ter* migration to midcell is unclear, on arrival it is partially anchored to the divisome by a protein linkage involving FtsZ, ZapA, ZapB and MatP[22]. Disrupting this linkage has previously been shown to reduce the duration of *ter* centralisation and alter the timing of sister *ter* segregation[19,22,29]. However, when we imaged a *zapB* deletion strain (Supplementary Fig. 6), we found that the effect on MatP foci positioning was relatively minor, with only a

slight broadening of its position distribution and slightly earlier segregation (compare Supplementary Fig. 6, Fig. 1). The transition to midcell still occurs rapidly (Fig. 5a, b). Interestingly, while *ori* duplication occurs at approximately the same time (Supplementary Fig. 6e), the asymmetry in its segregation is almost absent with only a small difference in the velocity of sister *ori* detectable (Fig. 5c, d). Nevertheless, *ter* centralisation is still coincident with the completion of origin segregation (Fig. 5b, d).

A stronger phenotype was obtained in *matPΔC20* cells, in which MatP lacks 20 amino acids from its C-terminal, which is believed to prevent its multimerisation and interaction with ZapB but not *matS* binding[17,19,21,22]. While we again found that the timing of *ter* centralisation is very similar to wildtype, *ter* is more dynamic and often overshoots the mid-cell before returning (Supplementary Figs. 7a, 5c). This was apparent at the population level as a smear in the kymograph (Supplementary Fig. 7c) and is in agreement with previous work[29]. The segregation of sister *ter* occurs noticeably earlier and as a result there is a correspondingly shorter period of centralisation (Supplementary Fig. 7c, e). Despite these abnormalities, *ori* positioning seems largely unaffected (Supplementary Fig. 7b, f) and importantly the completion of *ori* segregation remained coincident with *ter* centralisation (Supplementary Fig. 7e, f, h, i). Overall, these data confirm earlier results that the migration of *ter* to mid-cell, and its maintenance there, does not require the Ter-linkage, while at the same time showing the robustness of the coupling between *ter* centralisation and completion of *ori* segregation.

### *ori* segregation is a requirement for stable MatP relocalization to mid-cell

We had more success disrupting *ori* segregation. The original ParB$_{P1}$ labelling system produces foci numbers consistent with measurements of DNA content when used at basal expression levels, i.e. uninduced[13,16]. However, high levels of induction result in a segregation defect with fewer and larger ParB foci detected[16]. This was an initial challenge as continuous imaging in the mother machine requires sufficiently high expression and continuous induction. However, we found that the defects were attributable to the dimerisation of the CFP fluorophore used since no defects were detectable when using the

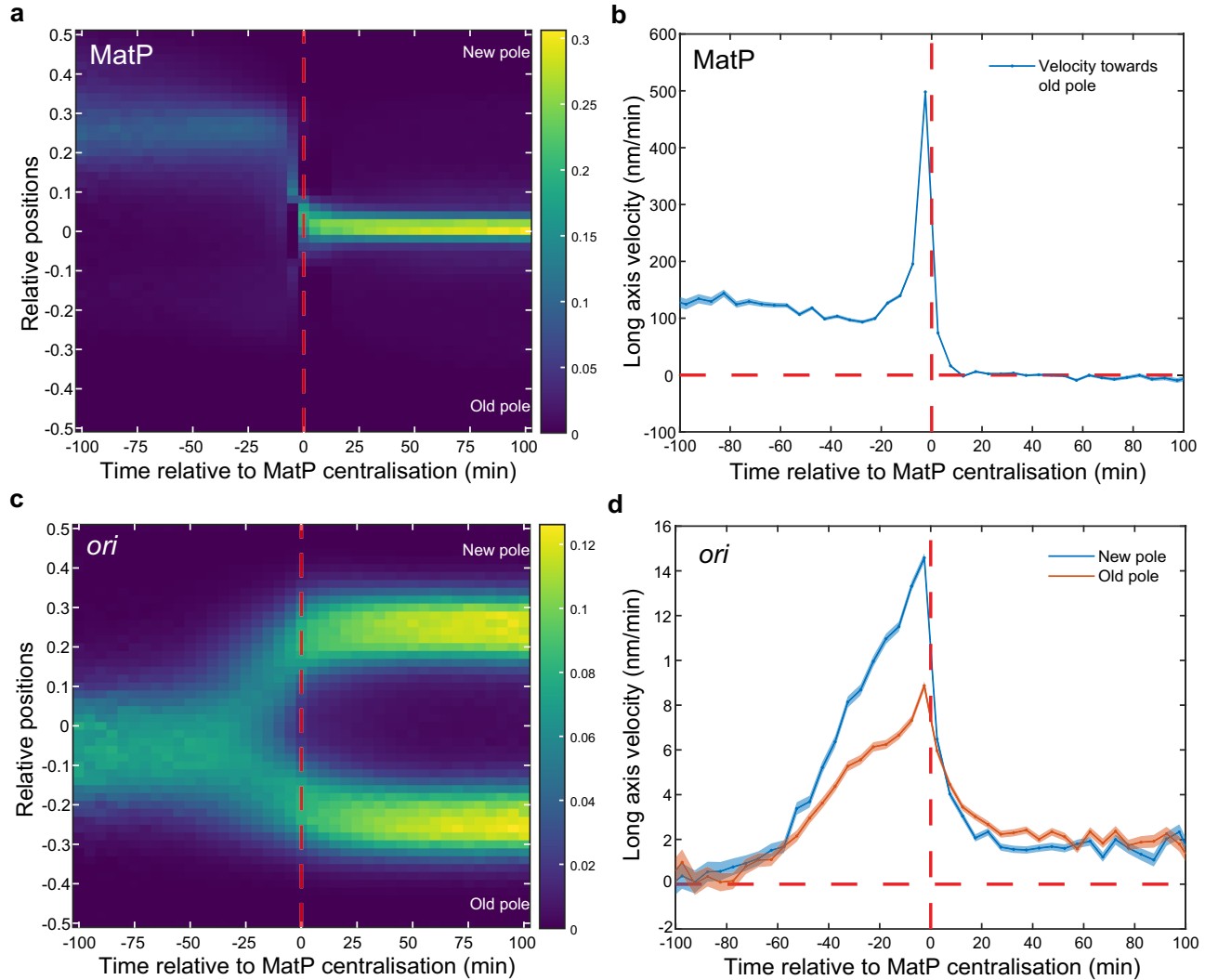

**Fig. 4 | *ter* centralisation coincides with completion of *ori* segregation.**
**a** Kymograph of MatP foci positions synchronised to MatP centralisation. **b** Mean velocity of the MatP focus towards the old pole relative to the time of MatP centralisation. **c** Kymograph of *ori* foci positions synchronised to MatP centralisation

**d** Mean velocity of *ori* foci towards the nearest pole relative to the time of MatP centralisation. Data as in Fig. 1. The colour scales in **a** and **c** are as in Fig. 1. Shading in (**b**, **d**) indicates standard error of the mean. Source data are provided as a source data file.

monomeric mTurquoise2 fusion to visualise *ori*, presented thus far (see methods). Nevertheless, this finding was fortuitous as we could use the induced CFP-ParB$_{P1}$ system as a tool to disrupt *ori* segregation. This has advantages over, for example, depleting TopoIV[49] as the direct effect of the perturbation should be local to the *ori* region.

We found that more than half of the cells with induced expression of CFP-ParB$_{P1}$ displayed the same spatiotemporal organisation of *ori* and *ter* at division as seen previously for the mTurquoise2-ParB$_{P1}$ labelling system (Fig. 6a, top). However, the remaining ~46% showed a defect in *ori* segregation, with only a single mid-cell localised CFP-ParB$_{P1}$ (*ori*) focus visible for most of the cell cycle (Fig. 6a, middle and bottom). At the same time, *ter* (MatP-YPet) does not maintain a sustained mid-cell localisation as in normal cells. It still moves to mid-cell but only for a short time, likely in order to be replicated, as evidenced by the appearance of two foci shortly afterwards. These MatP foci were often found to rapidly move outward towards opposite poles, resulting in both daughter cells having an inverted *ori-ter* axis i.e. with *ter* at the old rather than new poles (Fig. 6a middle, 6b, Supplementary Fig. 9a). Alternatively, one chromosome manages to correct itself before division resulting in only one daughter cell having an inverted orientation (Fig. 6a bottom, 6b, Supplementary Fig. 9b). Inverted

daughters grew normally and showed comparable growth rates and cell cycle durations (Supplementary Figs. 8e, f and 9a, b), indicating successful inheritance of the chromosome from the parent. This shows that while a new-pole-oriented *ter* is the norm, neither it nor *ori* segregation away from mid-cell, are requirements for successful completion of the cell cycle. Indeed, in our data sets in Figs. 1, 2, we also find cells with an inverted orientation, albeit at a very low frequency of 0.5–1% (examples from the nucleoid-labelled strain of Fig. 2 are shown in Supplementary Fig. 9c, d). We also do not observe a hereditary effect on orientation - cells born with an inverted orientation have a similar probability to produce inverted daughters as cells born with a normal orientation (Supplementary Fig. 8c).

This mis-positioning of *ori* and *ter* are clearly reflected in the averaged kymographs, which contrast strongly with the normal population (Fig. 6c, Supplementary Fig. 8a, b). Note that while *ter* shows some period of mid-cell localisation in these kymographs, it is a result of variation between cells: It spends on average one-third of the time at mid cell compared to the normal subpopulation or to cells with *ori* labelled using mTurquiouse2-ParB$_{P1}$ (Fig. 6d). Colocalisation with *ori* is also not increased compared to normal cells (Supplementary Fig. 8d). Overall, these results suggest that segregation of

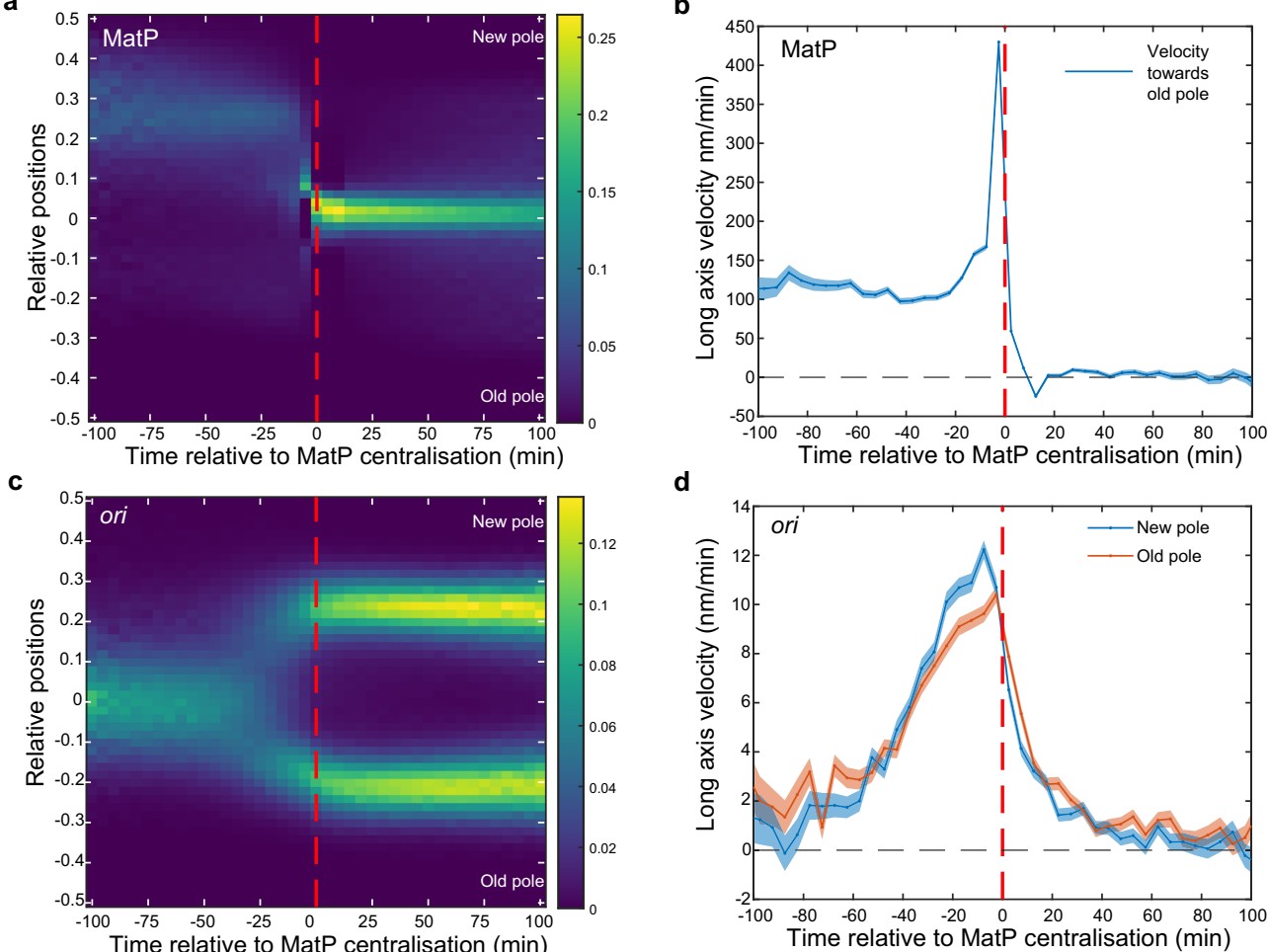

**Fig. 5 | *ter*-linkage is not involved in MatP relocalization. a** Kymograph of MatP foci positions in a *zapB* strain synchronised to MatP centralisation. **b** Mean step-wise velocity of MatP focus tracks towards the old pole relative to MatP centralisation in a *zapB* strain. **c** Kymograph of *ori* foci positions synchronised to MatP centralisation in a *zapB* strain. **d** Mean velocity of *ori* foci tracks towards the nearest pole relative to the time of MatP centralisation in a *zapB* strain. *n* = 38071 cycles. Shading in (**b**) and (**d**) indicate standard error of the mean. The colour scales in (**a**) and (**c**) are as in Fig. 1. Source data are provided as a source data file.

sister *ori* away from mid cell is required before *ter* can be stably localised to that position.

We also attempted to examine the effect of decreased *ori* cohesion by mildly over-expressing TopoIV using an arabinose inducible promoter[21,50]. However, for unknown reasons, we found the opposite behaviour: *ori* cohesion appeared to increase as evidenced by a delay in *ori* segregation and the presence of bright *ori* foci (Supplementary Fig. 10). Nevertheless, MatP centralisation remained coincident with the completion of *ori* segregation (Supplementary Fig. 10c, d).

What might underlie this apparent repulsion between *ori* and *ter*? The Structural Maintenance of Chromosomes complex MukBEF is an important chromosomal organiser in *E. coli* and is required for correct *ori* positioning[51–53]. It forms DNA-associated clusters that colocalise with *ori* but is displaced from the terminus region by its interaction with MatP[19,20], making it a plausible mediator of the coupling we observe. However, we found that *ter* positioning is largely unchanged in the absence of MukBEF (Supplementary Fig. 11a) consistent with a recent study[40]. Importantly, it still displays a similar rapid transition from the new pole to midcell. Unfortunately, our *ori* labelling system (mTurquoise2-ParB$_{P1}$) appeared to induce defects in the absence of MukBEF as evidenced by a large increase in anucleate cells (occurring in approximately 25% of divisions) and a somewhat disrupted *ter* positioning when we additionally labelled *ori* (Supplementary Fig. 11b, c). It may be that TopoIV recruitment by MukBEF is required to

counteract supercoiling induced by ParB[54]. We therefore cannot determine if there is a difference in the relative timing of *ter* centralisation and *ori* segregation. However, the snapshot demographs of Mäkelä et al. based on FROS labelling of loci, though more limited in their resolution, appear to show that *ori* segregation from mid-cell to the poles is still roughly coincident with *ter* centralisation in the absence of MukBEF. While further work is required, these data suggest that MukBEF is likely not responsible for the coupling we observe. Note that we have no reason to suspect artefacts from the mTurquoise2-ParB$_{P1}$ labelling system in the other strains studied. The foci distributions we obtain are consistent with previous studies and no phenotypic changes were observed when a lower induction level was used.

## Discussion

### Stable *ter* centralisation is coupled to, and requires, *ori* segregation

Previous work based on fluorescence in-situ hybridisation (FISH) defined several organisational transitions of the *E. coli* cycle during slow growth[34,55]. The T1 transition is defined by the initial separation of sister *ori* at mid-cell. One sister moves to the *ter*-distal end of the nucleoid while the other remains close to mid-cell. The T2 transition occurs ~20 min later and marks the rapid movement of the remaining mid-cell-proximal *ori* to the *ter*-proximal edge of the nucleoid. This is also coincident with the movement of *ter* to mid-cell. The result is that

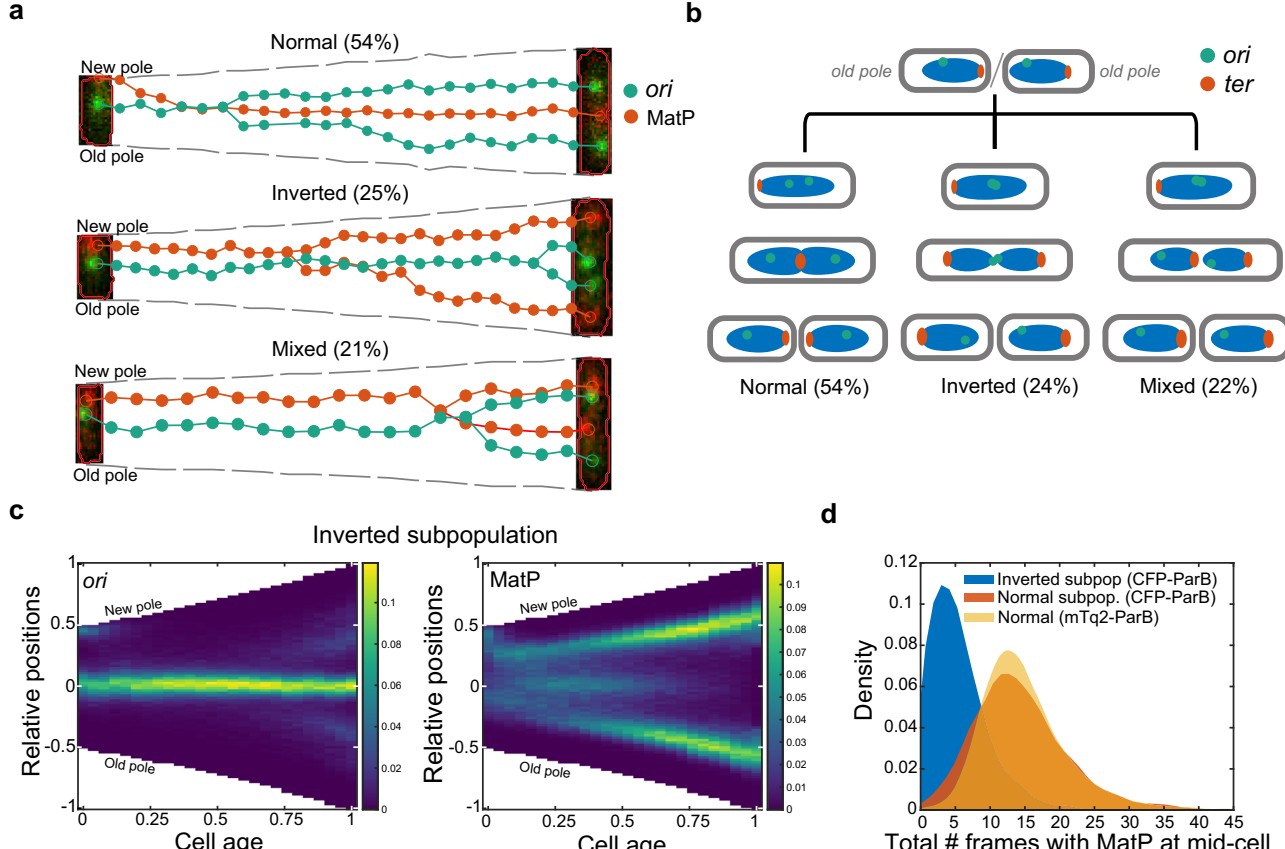

**Fig. 6 | Stable MatP localisation at mid-cell requires *ori* segregation. a** Example cell cycles beginning with a normal chromosome orientation but with daughter cells inheriting either normal, inverted or mixed (one normal, one inverted) orientations and their percentage occurrence within the normal-at-birth sub-population (see also Supplementary Figs. 8c, 9a, b). Foci position of *ori* (green) and MatP (red) are shown with a composite fluorescent image of the first and last frame of the cell cycle. **b** A schematic representation of *ori-ter* organisation based on (**a**)

and the previous nucleoid imaging (see also Supplemental Fig. 9). **c** Foci position kymographs of the inverted subpopulation (n = 3774 cycles). See also Supplementary Fig. 8a, b. **d** Distribution of the total number of frames in which MatP foci is found at mid-cell in each cell cycle of the inverted subpopulation (*ori*-CFP-ParB, n = 3774 cell cycles), normal subpopulation (*ori*-CFP-ParB, n = 8269 cell cycles) and for the data from Fig. 1 (*ori*-mTq2-ParB, n = 38066 cell cycles). The colour scale in (**c**) is as in Fig. 1. Source data are provided as a source data file.

the sister *ori* are positioned symmetrically at opposite ends of the nucleoid which correspond to the quarter positions of the cell. Our results are in broad agreement with this study. However, while we do also observe an asymmetry in *ori* positioning and movement, our data show both sister *ori* reaching their target positions at the edge of the nucleoid simultaneously. Our tracking of tens of thousands of cell cycles at a higher temporal resolution also allows us to establish that *ter* centralisation is much more rapid than *ori* movement, occurring in about 5 min (1 frame) and more specifically is coincident with completion of the more gradual process of *ori* segregation.

What might underlie such a rapid transition? We find it unlikely that it is due to the action of the typically mid-cell localised replisomes 'pulling in' the *ter* region given the large variation in the time between initial separation of *ori* foci and *ter* centralisation (Fig. 1c). The same conclusion was made in a study of chromosome organisation during fast growth[46], which found that the transition can take place at very different stages in the replication process but that whenever it occurs any unreplicated DNA is brought with the terminus to mid-cell. This was attributed to the entropic properties of a replicating ring polymer. Indeed, polymer simulations have shown that a partially replicated chromosome in a rod-shaped cell will organise itself to place the unreplicated terminus region at mid-cell[56,57] and this can lead to rapid movement of the terminus region from the pole to mid-cell (on the timescale of the stochastic fluctuations in the position of the terminus region)[29].

Our results are consistent with this picture. We see a clear connection between origin segregation and *ter* centralisation, consistent with an entropically-induced global chromosome rearrangement[34,55]. While an *ori*-specific mechanism is not discounted, our finding that *ori* segregation is not a strict requirement for bulk chromosome segregation (see below) also supports a locus-agnostic process. Future polymer modelling studies should be able to clarify whether the abruptness and relative timing of the *ter* transition (coincident with the completion of *ori* segregation) that we observe could indeed have an entropic origin. Note that under this explanation, there is no direct causal relationship between *ter* centralisation and the completion of *ori* segregation: both processes occur concomitantly as part of an entropically-driven rearrangement of the replicating DNA polymer.

## *ori* segregation is not required for bulk chromosome segregation

In *E. coli* and other studied bacteria, chromosome segregation begins with the origin and proceeds progressively as the chromosome is replicated. Indeed, the importance of initial origin segregation is underlined by the presence, in many species, of a dedicated system (ParABS) responsible for this task. It was therefore surprising to find that the origin segregation defect occurring in roughly half of cells using the CFP-ParB_{P1} labelling system (and ~0.5% of cells using mTurquoise2-ParB_{P1}), did not prevent or inhibit successful completion of the cell cycle (Fig. 6). These cells maintained unsegregated *ori* at

mid-cell for the majority of cell cycle but nevertheless segregated their chromosomes into each cell half, producing daughter cells with an inverted chromosome orientation i.e. *ter* at the old and *ori* at the new pole. This result indicates that while origin segregation typically leads the process of chromosome segregation, it is not a requirement for it to do so, consistent with entropy being the underlying driver of segregation. Note that since we label one specific locus (14 kb from *oriC*), it is possible that the rest of the Ori macrodomain segregates properly with the *parS* proximal regions being maintained together at mid-cell. However, this would not explain the maintenance of the inverted pattern both after the sister *ori* finally segregate and in the daughter cells of the next generation.

### The organisation of the chromosome

Two different organisational patterns of the chromosome of slow-growing *E. coli* have been described. The earlier view is based on FISH analysis of *ori* and *ter* and nucleoid labelling[34,35,38]. The chromosome is initially organised longitudinally with *ori* and *ter* at opposite extremes of the nucleoid. The *ori* moves towards mid-cell/nucleoid where it is replicated. Sister *ori* then migrate outwards to opposite edges of the nucleoid, which correspond to the quarter positions of the cell. A similar pattern occurs in *Bacillus subtilis*[58]. In the second view, based primarily on snapshot imaging of live cells[13,14,16,30], the chromosome initially adopts a lengthwise ('transverse') left-*ori*-right orientation with *ori* at mid-cell and the two chromosomal arms on either side connected by a stretched terminus region. During chromosome replication, the sister *ori* segregate to the cell quarters and the left-*ori*-right pattern is reproduced in each cell half, ensuring inheritance by the daughter cells. Subsequent studies from other groups have reproduced these results[10,41] and this transverse (left-*ori*-right or 'sausage') model has become the accepted picture of chromosome organisation in *E. coli* during slow growth. However, longitudinal organisation is still relevant as two more recent live-cell studies have found its appearance at faster growth (-1 h doubling time) with and without overlapping replication[45,46].

Indeed, our observations of *ori*, *ter*, left and right loci are most consistent with this pattern and indicate that a longitudinal organisation can also occur during slow growth. However, the *ori* are not located at the extreme edge of the nucleoid but close to the outer quarter positions i.e. 25% of the nucleoid mass lies between each *ori* and the closest pole. Thus we refer to it as a mixed or longitudinal-like organisation. Note that we have no reason to doubt the previous studies supporting a transverse pattern. The question is then: what determines which organisational pattern is observed? The growth media is an obvious possibility. However, we found a similar positioning of *ori* at the quarter position of the nucleoid mass at both the beginning and end of the cell cycle when cells were grown with glycerol rather than glucose (Supplementary Fig. 12) as well as in AB media as was used in previous studies of the same MG1655 wildtype strain[13,16], notwithstanding that under these conditions the *ori* displays a prolonged positioning at mid-cell/-nucleoid prior to its duplication. We therefore speculate that the observed differences are due to the differing conditions of our approach. While in our mother machines cells are maintained in steady state growth (Supplementary Fig. 1d), this may not be the case on agarose pads, especially using minimal media. The constant flow of fresh liquid media through the device may also mean that the conditions are more similar to batch liquid culture and cells most likely grow faster than they would on agarose pads. While further study is required, our work nevertheless shows a longitudinal chromosome organisation can occur during slow growth. In this regard, *E. coli* may share some similarity with *Bacillus subtilis* which switches between both patterns during its vegetative cell cycle[58].

### The role of the *ter*-linkage

The *ter* region is believed to be anchored to the septal ring through a MatP-ZapB-ZapA-FtsZ protein linkage. Disruption of this linkage was found to reduce the duration of, but not entirely abrogate, the mid-cell localisation of *ter* and leads to earlier separation of sister loci[22,29]. However, while we indeed observed this for the *matPΔC20* strain (Supplementary Fig. 7), deletion of *zapB* had a very mild effect on *ter* centralisation, with only a slightly wider position distribution and slightly earlier release of sister *ter* detectable (Supplementary Fig. 6). This suggests that MatP may interact with other components of the divisome through the same C-terminal domain or that some part of its anchoring effect requires its C-terminal domain independently of its linkage to the divisome. The latter could be connected to MatP multimerization and bridging of sister *ter*[17,18].

### Quantitative timings of cell cycle events

Our high throughput timelapse approach allows the determination of the entire distribution of steady-state cell cycle event timings and lengths. In particular, we measure a median time of 30 min between *ori* focus duplication and *ter* centralisation and 45 min between *ter* centralisation and stable nucleoid constriction, with these events occurring in this order in 87% of cell cycles (Supplementary Fig. 4d). These results complement recent studies on the relative timing of replication termination and the onset of cell constriction[59,60] and contribute to our understanding of cell cycle progression. The large size of our dataset (tens of thousands of cell cycles) also allows us to quantify the substantial variation between cells.

Interestingly, we found that 15% of cells were born with two *ori* foci, the majority of which do not initiate replication within their cell cycle (Supplementary Fig. 1f). This occurs due to the initiation of a second replication (C period) late in the cycle of the mother cell that then continues into the daughter cells. While this observation cautions against the textbook picture of one initiation per cell cycle, studying these 'outlier' cell cycles may be informative for the study of cell cycle control[32,61].

## Methods

### Strains, plasmids and media

All strains used in this study are derivatives of *E. coli* MG1655. For imaging origin and terminus, a new strain was constructed by transduction of *glmS::parS_{P1}::kan* from strain RM29[18,28] and *matP-YPet::frt::kan::frt* from strain RH3[29] into MG1655 (lab collection) respectively. Two plasmids - pFHCP1-CFP and pFHCP1-mTurquoise2, derived from plasmid pFHC2973[13] by deletion of *ygfp-parB_{pMT1}*, were used to drive the expression of CFP-ParB_{P1} and mTurquoise2-ParB_{P1} respectively to visualise *ori*. Additionally, a third plasmid was also constructed, namely pFHCP1-mTurquoise2-T1-mVenus, derived from pFHC2973 by replacement of *cfp* and *ygfp* with *mTurquoise2* and *mVenus*. This plasmid was used to drive the expression of mTurquoise2-ParB_{P1} and mVenus-ParB_{pMT1} to allow for simultaneous visualisation of *parS_{P1}* and *parS_{pMT1}* tagged chromosomal loci respectively. The plasmid pFHCP1-mTurquoise2-TopoIV, used to label *ori* as well as overexpress TopoIV, was constructed by amplifying the *araC::P_{BAD}::parE::parC* region from pWX35 plasmid and cloning into pFHCP1-mTurquoise2 plasmid. To create the triple labelled strain IS 129, strain RH3 was transduced with *glmS::parS_{P1}::kan* after removal of the kanamycin resistance. *parS* sites at *elaD* and *rhlE* loci were inserted by amplifying *parS_{P1}* or *parS_{pMT1}* sites from plasmids pGBKD3-parSP1 and pGBKD3-parSpmT1 respectively[18,62]. A detailed list of strains and plasmids used in this study can be found in Supplementary Tables 1 and 2. All experiments, unless otherwise mentioned, were performed at 30°C using M9 minimal media (1x M9 salts supplemented with 0.2% glucose, 2 mM MgSO_4, 0.1 mM CaCl_2). For experiments involving AB minimal media, the recipe and growth conditions of ref. 13. were followed. For the TopoIV overexpression experiment, glucose was substituted with glycerol and 0.0005% arabinose. For all the mother machine experiments, media were supplemented with 0.5 mg/mL BSA (for passivation to reduce cell adhesion to PDMS) and 50 μM IPTG (for

induction of mTurquoise2-ParB$_{P1}$ or CFP-ParB$_{P1}$, or mTurquoise2-ParB$_{P1}$ and mVenus-ParB$_{pMT1}$). Unless otherwise mentioned, the growth media or the overnight culture do not contain antibiotics.

## Microscopy

Strains were grown overnight in the respective minimal media without BSA and IPTG. For induction of mTurquoise2-ParB$_{P1}$, CFP-ParB$_{P1}$ or mTurquoise2-ParB$_{P1}$ and mVenus-ParB$_{pMT1}$, 50 μM IPTG was added to cultures 3 h before loading into the mother machine microfluidics device. The microfluidics device was prepared and loaded as described previously[26]. The cells in the exponential phase were then loaded into the mother machine using a 1 mL syringe. After loading, cells were fed with fresh M9 minimal media supplemented with 0.5 mg/mL BSA and 50 μM IPTG at a rate of 2 μL/min, and data were acquired after 3 h. Time-lapse images were taken every 5 min using a Nikon Eclipse Ti-E with a 100x oil-immersion objective and a ORCA-Fusion C14440-20UP camera (Hamamatsu photonics) with a pixel size of 0.065 μm (at 100x magnification) using Nikon NIS Elements AR 5.20.01. Both phase contrast and fluorescence signals were captured as mentioned above for up to 72 h. Visualisation of mTurquoise2-ParB$_{P1}$ required blue light excitation (wavelength 436 ± 20 nm), which is known to cause cell cycle arrest[63]. We optimised our imaging settings to avoid this and allow sustained imaging over several days. With these settings, cells divided on average 13% later but we saw no evidence of cell cycle arrest or other defects. The same imaging settings were used for all strains in this study (even for strain IS 173, in which mTurquoise2-ParB$_{P1}$ is not present). The IPTG concentration used for induction of mTurquoise2-ParB$_{P1}$ or CFP-ParB$_{P1}$ did not result in a change in growth rate or cell cycle duration under our imaging conditions.

## Image analysis

All analyses were performed using MATLAB 2020b. Time-lapse microscopy images were analysed using our custom-built pipeline called Mothersegger[26,33]. Briefly, time-lapse images acquired using the method described above were saved as TIFF stacks. The pipeline then identifies and isolates individual growth channels, performs background subtraction, and runs segmentation. The segmented data are used to identify cells belonging to the same cell cycle, along with their parent and daughters. A hard cut-off was put in place to discard cell cycles that are less than 10 frames (50 min) and greater than 60 frames (300 min) for experiments in glucose media. For experiments in glycerol media, the upper limit was raised to 80 frames (400 min). After identifying cell cycles, foci detection is performed on relevant fluorescence channels.

An important step in our image analysis pipeline is the correction of any lateral offset (pixel shift) between the phase contrast and fluorescence channels, especially in the vertical (long cell axis) direction. We have observed offsets of up to ~2 pixels that can introduce substantial noise into the position distribution (kymographs/demographs) if not corrected (note that organising cells by polarity results in two populations with opposite offsets, thereby compounding the effect). Since the offsets are not consistent between imaging sessions (and hence are not entirely due to the optical properties of the filters etc.), they must be determined for each data set separately. This is done by realising that, in the absence of an offset, averaging the fluorescent line profile from cells with opposite polar orientations in the device should lead to a symmetric profile irrespective of any asymmetry within individual cells. We therefore find the corrective offset that symmetrises this profile.

## Data analysis

While we have focused on time-based measurements, given that there is a known dependence of chromosome replication initiation on cell size[32], we also considered the cell length at which cell cycle events occurred (Supplementary Figs. 1 and 3). However, comparing kymographs (based on relative cell age) and demographs (based on cell length) we saw no indication that cell size is a better metric for studying *ter* centralisation. On the other hand, since *ter* centralisation was proposed to be coordinated with its replication[22], we reasoned that a time-based analysis would be most appropriate.

Measurements of growth rate are performed at the level of individual cell cycles by fitting the cell area on each frame to an exponential function.

## *ori* duplication

To define the *ori* focus duplication, we analysed complete cell cycles to identify the first frame in which two *ori* foci were detected for the first time. For cells that have an *ori* focus duplication recorded before the first 4 frames, we go back to its mother cell to identify or confirm the correct timing of *ori* focus duplication. In cases where the duplication is recorded in the mother cell, a negative frame number is recorded for *ori* duplication for the daughter cell.

## MatP relocalization

We defined MatP centralisation time point as the first of the three consecutive frames (15 min) in which MatP is seen at the mid-cell for the first time in the cell cycle. The mid-cell region is defined as the central 4.8 pixels (0.32 μm) in each cell. This value is identified by analysing the spread of MatP foci positions in the demograph (Supplementary Fig. 1h) when it is tightly localised at mid-cell (between 2.48 and 3.01 μm long cells). The duration of three consecutive frames was chosen as a compromise between a time frame long enough to filter out transient *ter* centralisation events while being short enough to minimise the effect of missed foci, in particular for Δ*zapB* and *matP*Δ*C20* strains.

## Nucleoid constriction and HU contours

The line profiles of HU-mCherry are obtained by taking the mean of the signal along the short axis. A constricted nucleoid is defined by a dip in the middle one third of the line profile greater than the threshold value. We define the time of stable constriction as the earliest frame from which the nucleoid remains constricted until the end of the cell cycle. The threshold (0.13) was chosen as the 95th percentile of the relative depth of the nucleoid signal in new-born cells i.e. the first bin of the plot in Supplementary Fig. 3b. This was done to account for noise in the nucleoid profile (no smoothening is performed) and was based on our observation that newborn cells do not have a constricted nucleoid. This method gave results consistent with manual inspection of the images.

The threshold intensity values defining the contour lines of the HU-mCherry kymographs were chosen such that that 50% or 80% of the total HU-mCherry signal is contained within the contours. The threshold is calculated for each time slice of the kymograph and then averaged.

## Reporting summary

Further information on research design is available in the Nature Portfolio Reporting Summary linked to this article.

## Data availability

All processed datasets generated in this study are available on the Edmond repository of the Max Planck Society at https://doi.org/10.17617/3.RMCWHK. Due to their large size, raw images are available upon request from the corresponding author. Source data are provided with this paper.

## Code availability

Image analysis was performed using the custom Matlab code 'Mothersegger'[26] found at https://gitlab.gwdg.de/murray-group/MotherSegger. Matlab code used for further analysis and figure

generation is available at https://gitlab.gwdg.de/murray-group/ori-ter-dynamics.

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

## Acknowledgements
The authors thank Frédéric Boccard, Olivier Espéli, Jaan Männik and François Cornet for strains and plasmid. This work was funded by the Max Planck Society.

## Author contributions
S.M.M. conceived the project. I.S. performed the experiments. I.S. and S.M.M. analysed the data and wrote the paper.

## Funding

## Competing interests
The authors declare no competing interests.
