## [Peer Review File · Nature Communications]

Mid-cell migration of the chromosomal terminus is coupled to origin segregation in *Escherichia coli*Reviewer #1 (Remarks to the Author):

After birth of slow growing *E. coli* cells, the chromosomal origin region (*ori*) is usually located near midcell and the terminus region (*ter*) near the newest cell pole. Somewhere after initiation of DNA replication, the sister *ori*'s then move apart to \sim quarter positions while the (as yet unreplicated) *ter* moves to midcell at some point.

Here, the authors explored the timing of, and requirements for, the *ter* centralization step in cells growing with a doubling time of \sim 130 min in a mother-machine by monitoring the sister *ori* segregation and *ter* centralization steps during thousands of cell/division cycles at a relatively high (5 min) temporal resolution.

The most notable results of these studies are that *ter* centralization is a rapid process that occurs well before visible segregation of nucleoids (nucleoid constriction) and, importantly, that it coincides with the completion of sister *ori* segregation. Moreover, its temporal coupling with completion of *ori* segregation is not dependent on a functional Ter-linkage with the division apparatus, and likely also not dependent on MukBEF. They further show that artificially increasing the time of sister *ori* 'cohesion' interferes with normal *ter* centralization, and can lead to the generation of viable cells with an inverted *ori-ter* axis. The data also lead to a refinement in the positioning of *ori* relative to the cell (slight bias towards cell poles) and the bulk nucleoid (also less central than currently thought). Overall, the results are consistent with models in which both bulk segregation of the chromosome and rapid *ter* centralization are entropy driven processes.

The quantity and quality of data and its presentation (writing and figures) is high overall, and the results are both interesting and important.

Critiques are that some of the authors assertions may be overstated, a possible involvement of FtsK in *ter* centralization is ignored, and that the accuracy/understanding of some text and figures can be improved.

Specific remarks

1) Line and page numbers would have rendered review of the manuscript more efficient.

2) Page 2, last sentence.

The authors assert that 'our results show that the mid-cell positioning of the terminus region is not due [to] its replication by mid-cell proximal replisomes or its linkage to the divisome'.

I have two comments:

a) As the authors neither study *oriC* cohesion nor replisomes in the manuscript, it is not clear how the results presented show that 'the mid-cell positioning of the terminus region is not due [to] its replication by mid-cell proximal replisomes'. If cohesion of newly replicated *oriC*'s were to exceed \sim 25 min or so under these specific growth conditions in the mother-machine, it is conceivable that replisomes indeed reached *ter* around the time of its centralization (i.e. \sim 25 min after initial separation of replicated *ori*'s, allowing for a C-period of \sim 50 min). So, please better explain to the reader what this assertion is based on.

b) The authors provide evidence that the Ter-linkage (FtsZ-ZapA-ZapB-MatP) between *ter* and the divisome is not required for *ter* centralization. However, the linkage between FtsK (divisome) and *ter* DNA was not examined. So, the latter part of the statement seems too broad as well.

3) Cell growth rates, focus velocities.

a) What 'per minute' units for growth rates are used for Fig. S1c,d, S2c, and S7e? Should this be micrometer per minute? If so, why not use a nanometer per minute scale? The multipliers at the top of the Y-axes are easy to miss.

b) Similarly, the multipliers in the focus velocity panels (Fig.4b,d, 5b,d, S6h,i) are hard to see, so I'd recommend a nanometer per minute scale.

c) What are 'growth rates for cell cycles' in the legend to Fig.S1c?

4) Fig. S1k, S3e

Please define PDF (Poisson/position distribution function/fraction?) in the legends.

5) Page 4. 'separation of ter foci occurred just before cell division'

What do you mean with 'cell division'? The onset of constriction, or cell separation (birth)? These events are likely at least 20 min apart under these growth conditions.

6) Page 9. The authors note 'that the time between ori duplication and ter centralization is decreased in both Ter-linkage mutants compared to wild-type', but this is difficult to infer from the figures.

a) How much shorter? Perhaps a table with relevant numbers can be added?

b) Do ori and/or ter foci move faster, or is the period between ori duplication and ori movement shortened?

c) Why would a defective Ter-linkage have this effect?

7) Page 9. The authors conclude ' that the migration of ter to mid-cell, and its maintenance there, does not require the linkage of the terminus region to the divisome '. However, they ignore the fact that, besides the Ter-linkage, FtsK also interact with the terminus region. Though FtsK activity may partly depend on the Ter-linkage, I don't believe it has been ruled out that FtsK can still bind ter, and perhaps help direct its location, in the absence of a functional Ter-linkage. If it has, please enlighten the reader.

a) So, this statement seems too broad, and I'd recommend rewriting it (see also point 2 above).

b) It also raises the interesting and relevant question if FtsK is then perhaps involved in ter centralization in wt and/or Ter-linkage mutant cells. Have the authors performed similar experiments with appropriate ftsK mutants (e.g. lacking the FtsK C-terminus)?

8) Pages 10-11. The authors nicely show that artificially increasing the time of sister ori 'cohesion' interferes with normal ter centralization. A converse experiment would be to decrease the sister ori cohesion period, which is observed when cells modestly overproduce TopoIV (Wang_G&D 2008). Would early completion of ori segregation also result in 'early' ter centralization, or would this cause temporal uncoupling of the events? Have the authors considered/tried this?

9) Methods

a) Strains RM29 and RH3 are missing from Table S1.

b) Table S2. Please provide a proper genotype of plasmids, including origin of replication, antibiotic resistance markers, and relevant (inducible) promoters and fused genes.

c) Page 15, medium. No antibiotic is mentioned. So, was antibiotic to select for parB plasmids (Amp?) left out of the growth medium in the pre-culture and the mother-machine?

10) Some other points

a) Page 7. You mean (Fig.1D, Fig.S1H) rather than (Fig.1C, Fig.S1G)?

b) Page 9, top. You mean (Fig. S6 E,F,H,I) rather than (Fig. 6 E,F,H,I)?

c) Legend to Fig. S5c. Incomplete sentence.

d) Legend to Fig. S6. Panels H and I are mislabeled G and H.

Reviewer #2 (Remarks to the Author):

The manuscript by Sadhir et al. describes a coordinated movement of the replication origin (oriC) and terminus (terC) region of E. coli chromosomes in slow-growth conditions. It is a careful quantitative study that elucidates the dynamics of chromosomes during the cell cycle. The work mostly revisits already-known findings but does it more quantitatively than the earlier studies. The main result, "Mid-cell migration of the chromosomal terminus is coupled to origin segregation in Escherichia coli" (the title of the manuscript), has been reported by Bates and Kleckner (Cell 2005): "Strikingly, concomitant with this transition, ter and the ter-side origin switch positions. [...] ter moves inward toward midcell while the ter-side origin becomes localized to the 1/4 position, symmetrical to its sister at the 3/4 position." The most interesting/novel finding from this work is that without splitting of oriC-proximal focus, the Ter still moves to midcell although only for short period of time. It is then instead localizing to the nucleoid periphery in cells where the ori-foci remain at the midcell. The motion to the midcell appears to be driven by DNA replication.

I would have expected some mechanistic insights or modeling arising from the data to be included.

Main points of criticism:

1) There is no mechanistic interpretation of findings, which mostly have been known before. Without model the impact of the work will be modest.

2) Title and the section "ter centralisation is coupled to, and requires, ori segregation" in Discussion. When ori region is crosslinked by CFP-ParB, ter centralization still occurs, although for a short period. It seems to me to contradict the statement that ori segregation is required for the ter centralization.

3) The aligned and averaged kymograph in Fig. 4 show vary rapid movement of the Ter focus. Such representation overlooks that Ter can frequently overshoot midcell position (Youngren et al GenesDev 2014), that the focus can split and that in individual cells the movement of Ter focus across the nucleoid can be observed. In other words, the movement of the Ter from the new pole side to nucleoid middle takes much more time then this kymograph shows. The alignment procedure in Fig. 4 is somewhat arbitrary. Why needs the Ter focus to be stable for three consecutive frames? What is the outcome if different criteria are used?

Minor points of criticism:

1) "While in some bacteria this can be directly attributed to the well-studied ParABS partitioning system, in other species this system is either not strictly essential or is absent altogether." – Some references needed.

2) "Here, we use high-throughput single-cell imaging and analysis to quantitatively establish the choreography.." choreography -> movement

3) "Indeed, and surprisingly given the slow growth rate, we found that 15% of cells were born with more than one ori focus (Fig. S1F), indicating that DNA replication was initiated in the previous cell cycle." – This should not be surprising. See S. Tiruvadi-Krishanan et al Cell Reports 2022 and P. Kar et al PNAS 2023.

4) "It was recently suggested that ter centralisation shortly precedes the stable appearance of a constricted/bilobed nucleoid structure (Männik et al., 2016)" – This is misquotation of Männik et al., 2016, which states "The permanent constrictions, which were present uninterrupted until cells divided, appeared in all strains only after the Ter region centralized (Figure 5C)." The transient nucleoid constrictions were seen before Ter centralization. Please correct the whole paragraph.

5) "Overall, these results confirm that the migration of ter to mid-cell, and its maintenance there, does not require the linkage of the terminus region to the divisome" – This was also shown in Mannik et al 2016.

6) Section "Quantitative timings of cell cycle events" – discussion why timings have been studied instead of cell lengths is more appropriate to the Results or Materials and Method section.

7) "Surprisingly, given the mean 133 min cell cycle duration, we found that 15% of cells were born with two ori foci, the majority of which do not initiate replication within their cell cycle (Fig. S1F)." –

The argument that the mean doubling time is 133 min in this growth condition is somewhat concerning. It might imply phototoxic effects. Some control with unlabeled strain is needed.

In this growth condition, the majority of cells should have two oris at birth. It is expected that because of noise in the initiation timing, a fraction of these cells "do not initiate replication within their cell cycle"

I would have expected some conclusion about the ori-ter dynamics rather than going tangentially to a discussion of double initiations.

Reviewer #3 (Remarks to the Author):

Overall impression

Sadhir and Murray present an interesting paper addressing the age-old question of how bacterial chromosomes segregate. Experiments are focused primarily on understanding segregation of the terminus region. This is understandable, as *ter* exhibits dramatic movements and is the only chromosomal locus (in *E. coli*) known to attach to cell structure capable of directing movement. Positions of *ori*, *ter*, and occasionally the nucleoid, are measured in slow growing cells by time lapse imaging in a microfluidics slide. Although there have been (many!) studies of *ori* and *ter* dynamics in twenty plus years, their approach is unique in that individual cells are examined over time with high temporal resolution. In addition, they avoid two pitfalls that have plagued many studies: dimeric fluorescent tags and unusual strain backgrounds. Analyses are sophisticated and adhere to a terminus-specific objective throughout the manuscript. They convincingly show that positioning of *ter* to midcell, an event presumed to be prerequisite for partitioning of daughter chromosomes at division, is independent of MatP-mediated linkage to the division apparatus but dependent on segregation of the origin region. Both of these findings are unexpected and important. Claims that origin segregation is only required for *ter* positioning (not the bulk nucleoid), and that chromosomes are oriented longitudinally and not transversely, are less convincing. Writing is crisp, and the authors have an uncanny knack for addressing questions and counter points at the moment they occur to the reader.

Major comments

1. Claims that delaying *oriC* segregation (using a dimeric CFP tag) does not disrupt bulk nucleoid segregation are unsupported by the data. While the effects on *ter* centralization are clear, there is no examination of nucleoid segregation per se. This is a major point in the paper, addressed in the results and discussion. Segregating nucleoids are even drawn in Figure 6B (although labelled "*ori*"), as if nucleoids were examined. HU or DAPI imaging should be performed to support this claim. Alternatively, other intermediate loci could be labelled, although this would be unnecessarily difficult unless the tools are already in hand. If possible, showing this data in supplemental would be useful to the field.
2. The longitudinal versus transverse debate (if there is one) is not answered by this paper. Like the early FISH studies proposing the longitudinal model with circular chromosome oriented sideways in the cell "*ori*-arms-*ter*", data in the current paper are restricted to *ori* and *ter*. The transverse model ("*left arm*-*ori*-*right arm*") on the other hand, was born from imaging several arm loci. The fact is, most data examining interstitial loci support the transverse data. I strongly advise softening anti-transverse-model statements in the discussion section "the organization of the chromosome", and instead focusing on how the current data confirms and extends the early *ori/ter* work and how it fits into the greater evolution of our view of *E. coli* chromosome organization.

Minor comments

1. Results 1st paragraph, please note that monomeric turquoise tagged ParB fusion was chosen to avoid artifacts of dimeric tags. It is widely known that the ParABS system was problematic, but not that it was due to tag multimerization. It should be mentioned here.
2. It is said that "thousands" of cells are imaged in experiments, but I cannot find *n* for most figures. Please add this information.
3. It is mentioned that variability was lower when relative cell ages are used instead of minutes after the previous division. If so, why isn't relative cell age used throughout? Example, Figures 3 and 4.
4. In the text relating to Figure 2, it is said that old/new pole bias of *ori* position cannot be determined from snap-shot analyses, however, this is not true (in fact it was in the early FISH papers). Polarity can easily be determined by position of *ter*, which is strongly new pole biased.
5. Figure 3A line colors are not explained.
6. First line in the results section "*ter* centralization coincides with completion of *ori* segregation" cites Figures 1C and S1G, but these are incorrect.
7. 2nd paragraph under "*ter-ori* coupling does not depend on the *ter* linkage. To my knowledge the

observation of ter overshooting the Z-ring in matPΔC20 strain is new and very interesting. Elaboration would be good here.

8. Same paragraph, "Fig 6 E,F,H,I" should be designated "S" supplemental.

9. Following paragraph, "consistent with flow cytometry" may not be understandable to many. Please add a brief explanation in the text.

10. Figure 6A, please add percentages for normal, inverted, and mixed classes (as in Fig S7C).

11. There is a large white space in the Methods section after "ori duplication". Is there missing text?

12. The distributions of cell cycle durations and growth rates seem wide, but it is difficult to compare to past studies. Comment? Ideally, you would want to compare yours to a study using batch cultures, not microfluidics.

Reviewer #1 (Remarks to the Author):

After birth of slow growing E. coli cells, the chromosomal origin region (ori) is usually located near midcell and the terminus region (ter) near the newest cell pole. Somewhere after initiation of DNA replication, the sister ori's then move apart to ~ quarter positions while the (as yet unreplicated) ter moves to midcell at some point.

Here, the authors explored the timing of, and requirements for, the ter centralization step in cells growing with a doubling time of ~ 130 min in a mother-machine by monitoring the sister ori segregation and ter centralization steps during thousands of cell/division cycles at a relatively high (5 min) temporal resolution.

The most notable results of these studies are that ter centralization is a rapid process that occurs well before visible segregation of nucleoids (nucleoid constriction) and, importantly, that it coincides with the completion of sister ori segregation. Moreover, its temporal coupling with completion of ori segregation is not dependent on a functional Ter-linkage with the division apparatus, and likely also not dependent on MukBEF. They further show that artificially increasing the time of sister ori 'cohesion' interferes with normal ter centralization, and can lead to the generation of viable cells with an inverted ori-ter axis. The data also lead to a refinement in the positioning of ori relative to the cell (slight bias towards cell poles) and the bulk nucleoid (also less central than currently thought). Overall, the results are consistent with models in which both bulk segregation of the chromosome and rapid ter centralization are entropy driven processes.

The quantity and quality of data and its presentation (writing and figures) is high overall, and the results are both interesting and important.

We thank this reviewer for their comments.

Critiques are that some of the authors assertions may be overstated, a possible involvement of FtsK in ter centralization is ignored, and that the accuracy/understanding of some text and figures can be improved.

Specific remarks

1) Line and page numbers would have rendered review of the manuscript more efficient.

We have added page numbers to the revised manuscript.

2) Page 2, last sentence.

The authors assert that 'our results show that the mid-cell positioning of the terminus region is not due [to] its replication by mid-cell proximal replisomes or its linkage to the divisome'.

I have two comments:

a) As the authors neither study oriC cohesion nor replisomes in the manuscript, it is not clear how the results presented show that 'the mid-cell positioning of the terminus region is not due [to] its replication by mid-cell proximal replisomes'. If cohesion of newly replicated oriC's were to exceed ~ 25 min or so under these specific growth conditions in the mother-machine, it is conceivable that replisomes indeed reached ter around the time of its centralization (i.e. ~ 25 min after initial separation of replicated ori's, allowing for a C-period of ~50 min). So, please better explain to the reader what this assertion is based on.

The replisome part of this statement is indeed out of place at this point in the manuscript (the introduction). It is discussed later in the results section and discussion and refers to previous work and the fact that we observe a very broad distribution in the time between ori duplication and ter centralisation. More specifically, the coefficient of variation (CV) of the C-period has been recently measured in not very different growth conditions at ~ 0.15 (Colin et al 2021, Elife), whereas we find that the time between ori duplication and ter centralisation has a $CV = sd/mean = 23.2/26.5 = 0.88$. The large difference in the variation of the two quantities is not consistent with centralisation being due to fork progression - we would expect a much tighter distribution if that was the case. However, we acknowledge that we do not directly show that fork progression is not responsible for the timing of ter centralisation. We have removed the comment from the introduction and now refer to the CV measurement above in the results section.

b) The authors provide evidence that the Ter-linkage (FtsZ-ZapA-ZapB-MatP) between ter and the divisome is not required for ter centralization. However, the linkage between FtsK (divisome) and ter DNA was not examined. So, the latter part of the statement seems too broad as well.

FtsK is required for the ordered *segregation* of the 200kb region surrounding the dif sites in the ter region at the end of the cell cycle and it requires MatP for this (Stouf et al. PNAS 2013, 10.1073/pnas.1304080110). To our knowledge, there is no evidence that FtsK is involved in the much earlier process of centralisation. Indeed, FtsK is only active shortly before and during septum constriction (ref 31, 32 of Stouf et al.) and centralisation is unchanged in an ftsK(ATP-) mutant (Galli et al, PLoS Genetics, 2017, 10.1371/journal.pgen.1006702).

Nevertheless, we now clarify that the statement refers specifically to the Ter-linkage.

3) Cell growth rates, focus velocities.

a) What 'per minute' units for growth rates are used for Fig. S1c,d, S2c, and S7e? Should this be micrometer per minute? If so, why not use a nanometer per minute scale? The multipliers at the top of the Y-axes are easy to miss.

The growth rate is the rate associated with the exponential growth of individual cells. For each cell cycle, it is found by fitting the cell area in time to an exponential. It has units of min^{-1} as indicated. This explanation was unfortunately only mentioned in the legend to Fig. S2 and not in S1. This has been corrected and also explained in the methods.

b) Similarly, the multipliers in the focus velocity panels (Fig.4b,d, 5b,d, S6h,i) are hard to see, so I'd recommend a nanometer per minute scale.

Done.

c) What are 'growth rates for cell cycles' in the legend to Fig.S1c?

They are the growth rates as above. The legend has now been rephrased. Note that if an exponentially-growing cell exactly doubles in size over the cell cycle then its cell cycle duration $= \log(2)/\text{growth rate}$. However, since cells do not exactly double in size (and can deviate from exponential growth), these two quantities are not identical.

4) Fig. S1k, S3e

Please define PDF (Poisson/position distribution function/fraction?) in the legends.

PDF (probability density function) has now been defined in the legends.

5) Page 4. 'separation of ter foci occurred just before cell division'

What do you mean with 'cell division'? The onset of constriction, or cell separation (birth)? These events are likely at least 20 min apart under these growth conditions.

It refers to the division of a cell into two daughter cells (we do not use the term separation as the daughter cells can stay in physical contact). Like all studies involving image analysis/segmentation, the assignment of the precise time-point of cell division is somewhat arbitrary and is chosen by the segmentation algorithm. The constriction of the cell typically begins to be visible in phase contrast 2-3 frames (10-15 min) before the division event is called. The parameters of the algorithm were chosen by hand to match our 'by eye' intuition. However the same parameters were used to analyse all the data sets in the study.

6) Page 9. The authors note 'that the time between ori duplication and ter centralization is decreased in both Ter-linkage mutants compared to wild-type', but this is difficult to infer from the figures.

a) How much shorter? Perhaps a table with relevant numbers can be added?

The violin plots are shown in Fig. S5E and S6G and the mean, sd and median values are given in the legends. However, the values given in the legend for matPC20 were incorrect. This has now been corrected. The timings in this strain are actually the same as WT. As we erroneously based our comment off these values, our statement was incorrect.

In zapB, the median time difference was given as 15 mins (mean+/-sd= 10.7 +/- 31.8 min) compared to 30 mins (26.5 ± 23.2 min) in the WT. However, we have now discovered that this is due to a higher rate of missed ori foci in that strain (for unknown reasons) (new Figure S6F). While this does not affect the kymographs, the quantification of the timing of ori duplication (the first frame on which two foci are detected) and hence the time difference are affected. When we restrict the analysis to cell cycles with <20% missed (false negative) foci (based on the tracking inferred using our *Track algorithm), we obtain a similar time for ori duplication (updated Fig. S6E), consistent with the kymographs. Thus we do not believe that there is a significant difference in the timings between either mutant and WT. However it is important to mention that Ter is more mobile in the matPC20 strain compared to WT. See the comment of Reviewer 2 below.

As a result of the above, the cited statement and related text have been corrected.

b) Do ori and/or ter foci move faster, or is the period between ori duplication and ori movement shortened?

The foci of both ori and ter do not appear to move significantly faster (compare Fig. 4B,D with Fig. 5BD).

c) Why would a defective Ter-linkage have this effect?

This comment is no longer relevant due to our response to point a).

7) Page 9. The authors conclude ' that the migration of *ter* to mid-cell, and its maintenance there, does not require the linkage of the terminus region to the divisome '. However, they ignore the fact that, besides the Ter-linkage, FtsK also interact with the terminus region. Though FtsK activity may partly depend on the Ter-linkage, I don't believe it has been ruled out that FtsK can still bind *ter*, and perhaps help direct its location, in the absence of a functional Ter-linkage. If it has, please enlighten the reader.

a) So, this statement seems too broad, and I'd recommend rewriting it (see also point 2 above).

b) It also raises the interesting and relevant question if FtsK is then perhaps involved in *ter* centralization in wt and/or Ter-linkage mutant cells. Have the authors performed similar experiments with appropriate *ftsK* mutants (e.g. lacking the FtsK C-terminus)?

As stated above, FtsK is only active just before division and is involved in *ter* segregation at the end of the cell cycle (as well its primary role in dimer resolution). There is no evidence that it has a role in the earlier process of centralisation.

In any case, the intent of our statement was that the Ter-linkage (FtsZ-ZapA-ZapB-MatP) is not required. We have now clarified this.

8) Pages 10-11. The authors nicely show that artificially increasing the time of sister *ori* 'cohesion' interferes with normal *ter* centralization. A converse experiment would be to decrease the sister *ori* cohesion period, which is observed when cells modestly overproduce TopoIV (Wang_G&D 2008). Would early completion of *ori* segregation also result in 'early' *ter* centralization, or would this cause temporal uncoupling of the events? Have the authors considered/tried this?

We are thankful for this excellent suggestion. We obtained the pWX35 plasmid used in Wang et al. 2008, from the Cornet Lab (Toulouse), who have used it in conjunction with the ParB labelling plasmid (Croizat et al 2020). We initially followed Croizat et al and examined cells containing both plasmids. However, we found that the presence of the same pBR322 origin on both resulted in increased heterogeneity in the mTurquoise2-ParB_{p1} signal. We therefore cloned the TopoIV genes under the P_{BAD} promoter from pWX35 into our *ori* labelling plasmid. However, using our growth conditions (which are similar to Croizat et al), we found no discernible difference in *ori* segregation/cohesion compared to WT. This was found to be attributable to strong glucose inhibition of the arabinose promoter (even at arabinose concentration of 0.2%). When we switched to glycerol as the carbon source, as in the original work by Wang et al, we could observe the toxic effects of TopoIV overexpression from as little as 0.001%. Though we were aware of the effect of catabolite repression by glucose on the arabinose promoter, the strength of the effect was unexpected, especially as Croizat et al claimed to have used the same construct successfully on M9 glucose.

In any case, we switched to media with glycerol as the carbon source and examined the effect of mild overexpression of TopoIV (using 0.0005% arabinose, as our combined plasmid appeared to have slightly lower expression, as measured by toxicity, than the original pWX35 plasmid). Contrary to previous reports, we found that cells, on average, have increased *ori* cohesion times. This was apparent in a delay in *ori* duplication timing (48 minutes and 77 minutes for WT and TopoIV overexpression respectively), despite comparable growth rate and division times. Though a delayed initiation of replication in TopoIV overexpressing cells cannot be ruled out, we found that some TopoIV overexpressing cells displayed a single, bright *ori* focus for most of the cell cycle, suggesting a duplicated, but cohesed *ori* (Fig. S10). Additionally, some TopoIV overexpressing cells appeared

elongated and occasionally produced anucleate cells, further suggesting an increase and not a decrease in *ori* cohesion. Note that our results are based on cells that were undergoing continuous induction which might have some global effects that lead to increased cohesion times which would not have been observable in short duration experiments as were the cases with most of the previous studies that investigated the effect of TopoIV overexpression. More work is required to understand the full effects of TopoIV over-expression.

Returning to the goal of perturbing the *ori-ter* coupling, we found that despite the abnormalities caused by TopoIV overexpression, *ter* centralization remained coincident with the completion of *ori* segregation (Fig. S10C, D). However we are hesitant to draw conclusions because of discrepancies in the reported phenotype of mild TopoIV overexpression and do not stress this result in the revised manuscript.

9) Methods

a) Strains RM29 and RH3 are missing from Table S1.

Added

b) Table S2. Please provide a proper genotype of plasmids, including origin of replication, antibiotic resistance markers, and relevant (inducible) promoters and fused genes.

Updated

c) Page 15, medium. No antibiotic is mentioned. So, was antibiotic to select for *parB* plasmids (Amp?) left out of the growth medium in the pre-culture and the mother-machine?

Yes. Preculture (M9-Glucose/Glycerol) and the growth medium (M9-Glucose/Glycerol+0.5 mg/mL BSA+50 μ M) in the mother machine did not include Ampicillin.

10) Some other points

a) Page 7. You mean (Fig.1D, Fig.S1H) rather than (Fig.1C, Fig.S1G)?

b) Page 9, top. You mean (Fig. S6 E,F,H,I) rather than (Fig. 6 E,F,H,I)?

c) Legend to Fig. S5c. Incomplete sentence.

d) Legend to Fig. S6. Panels H and I are mislabeled G and H.

Corrected

Reviewer #2 (Remarks to the Author):

The manuscript by Sadhir et al. describes a coordinated movement of the replication origin (oriC) and terminus (terC) region of E. coli chromosomes in slow-growth conditions. It is a careful quantitative study that elucidates the dynamics of chromosomes during the cell cycle. The work mostly revisits already-known findings but does it more quantitatively than the earlier studies. The main result, "Mid-cell migration of the chromosomal terminus is coupled to origin segregation in Escherichia coli" (the title of the manuscript), has been reported by Bates and Kleckner (Cell 2005): "Strikingly, concomitant with this transition, ter and the ter-side origin switch positions. [...] ter moves inward toward midcell while the ter-side origin becomes localized to the 1/4 position, symmetrical to its sister at the 3/4 position."

We are clear regarding our agreement with Bates and Kleckner. However, we point out that our high-throughput study is based on timelapse imaging of live cells at 5 min intervals, whereas Bates and Kleckner used FISH imaging of fixed cells taken from a synchronised culture at 15 min intervals. Therefore we can more precisely analyse the timing of the two events (quantifying the rapidness of ter centralisation and showing that it is coincident with *completion* of ori segregation) and we provide evidence that the relationship is not just a coincidence but rather that the two events are coupled. Furthermore the choreography of ori and ter during the cell cycle presented in Bates and Kleckner (ori at the edge of the nucleoid at birth etc) has been largely neglected in favour of the 'transverse' organisation of subsequent studies by the labs of Stuart Austin and Dave Sherratt and presented in almost all reviews since then (Wang et al, 2013 Nature Reviews Genetics; Wang & Rudner 2014 Curr. Opin. Micro.; Badrinarayanan et al 2015, Annu. Rev. Cell Dev. Biol.; Reyes-Lamothe & Sherratt. 2019 Nature Rev. Gen.). While the 'transverse' studies did image live cells (though primarily snapshot analyses), they generally did not simultaneously image the nucleoid (like Bates and Kleckner, Cell 2005). Thus, our study fills an important gap connecting these works and quantifies the dynamic and positioning of ori and ter relative to the nucleoid in live cells in unprecedented detail.

The most interesting/novel finding from this work is that without splitting of oriC-proximal focus, the Ter still moves to midcell although only for short period of time. It is then instead localizing to the nucleoid periphery in cells where the ori-foci remain at the midcell. The motion to the midcell appears to be driven by DNA replication.

I would have expected some mechanistic insights or modeling arising from the data to be included.

Respectfully, while modelling ori segregation and ter centralisation is a clear next step, it is reasonably left for future work. Polymer modelling of a replicating DNA polymer is a large endeavour, especially if the non-equilibrium effect of active replication is to be considered and we already have 6 main figures and 9 SI figures. Proper modelling, would also require the incorporation of DNA compaction and some scheme for ori segregation to, and maintenance at, the cell quarters, the mechanism for which is unknown.

Main points of criticism:

1) There is no mechanistic interpretation of findings, which mostly have been known before. Without model the impact of the work will be modest.

We provide an interpretation of our findings in the discussion section. We believe our results are consistent with the proposal of Youngren et al 2014 that the chromosome is a self-avoiding ring

polymer and that ter centralisation is driven by the polymeric/entropic effects acting on the replicating polymer. This comment also ignores the novelty of our results. Yes, chromosome organisation in *E. coli* has been studied for many years but we have made an important contribution here by our 1) high-throughput, steady-state quantification, and 2) identification and testing of a coupling between ter centralisation and the completion of ori segregation.

2) Title and the section “ter centralisation is coupled to, and requires, ori segregation” in Discussion. When ori region is crosslinked by CFP-ParB, ter centralization still occurs, although for a short period. It seems to me to contradict the statement that ori segregation is required for the ter centralization.

Like the title of the corresponding results section, the title should have been “*Stable* ter centralisation is coupled to, and requires, ori segregation”.

3) The aligned and averaged kymograph in Fig. 4 show vary rapid movement of the Ter focus. Such representation overlooks that Ter can frequently overshoot midcell position (Youngren et al GenesDev 2014), that the focus can split and that in individual cells the movement of Ter focus across the nucleoid can be observed. In other words, the movement of the Ter from the new pole side to nucleoid middle takes much more time then this kymograph shows.

Averaged kymographs must indeed be approached cautiously as the average picture may not necessarily be representative of the individual cells. While the position distribution is very narrow, individual tracks can still exhibit substantial mobility, just the average position is well defined.

We tried to be careful about this and referred to kymographs only as ‘indicating that the migration ... is relatively rapid compared to its movement at other times’. More importantly, this is the reason that we analyse the velocity of Ter in the individual cell cycles (panels B and D).

Importantly, we also showed that the frame with the largest ter movement coincides most frequently with the assigned time of stable centralisation, supporting the sharp transition in the aligned kymographs (we now also do the same for the two mutants (Fig. S5A-C)).

However, Ter over-shooting does happen in a minority of WT cells and in particular in the MatPC20 mutant, which we explicitly mention (previously noted in Männik et al 2016). However we do not understand the reference to Youngren et al. That is a study of fast over-lapping growth conditions, not the slow growth studied here and we can find no mention in Youngren et al of Ter overshooting midcell.

We now provide Ter tracks from a random selection of cells to provide the reader with a sense of the variability in Ter dynamics and centralisation (Fig S5A-C) and show the differences between the strains.

We also quantify the over-shooting etc by comparing the time of the first appearance of Ter at mid-cell with the time of stable centralisation (based on being at mid-cell for 3 consecutive frames) (Fig. S5G-I). These two times coincide 55% of the time for WT. In the other 45% Ter transiently enters the mid-cell region (leaving in either direction) at least one before being stable positioned there. In rare cases, this occurs very early in the cell cycle leading to a large difference in the two

times. Unsurprisingly the numbers are somewhat worse for the C20 mutant with agreement in ~40% of cell cycles.

The alignment procedure in Fig. 4 is somewhat arbitrary. Why needs the Ter focus to be stable for three consecutive frames? What is the outcome if different criteria are used?

A larger value would exclude a larger subset of the data (foci can be missing on some frames and so the data set is reduced the more consecutive frames are required; the data set would also be biased against cell cycles in which ter centralisation occurs late in the cycle and against short cell cycles). At 3 frames, we found that missed foci did not significantly affect the identification of the stable centralisation (blue bars in Fig. S5G-I). On the other hand, smaller values have a strong effect on the time of stable centralisation due to the transient centralisation events discussed above e.g. the mean time of first centralisation for the WT data set of Fig. 1 is ~40 min, compared to ~53 min for stable (3 consecutive frames) centralisation.

Thus the value of 3 frames is chosen as an arbitrary compromise between these two extremes. Importantly, we use the same value for all strains.

We have now expanded the methods section and main text to explain this.

Minor points of criticism:

1) “While in some bacteria this can be directly attributed to the well-studied ParABS partitioning system, in other species this system is either not strictly essential or is absent altogether.” – Some references needed.

References have been added.

2) “Here, we use high-throughput single-cell imaging and analysis to quantitatively establish the choreography..” choreography -> movement

Thank you for the suggestion but we prefer ‘choreography’ as it better refers to the positioning and movement of the loci relative to each other as well as to the cell.

3) “Indeed, and surprisingly given the slow growth rate, we found that 15% of cells were born with more than one ori focus (Fig. S1F), indicating that DNA replication was initiated in the previous cell cycle.” – This should not be surprising. See S. Tiruvadi-Krishanan et al Cell Reports 2022 and P. Kar et al PNAS 2023.

This appears to be a reference to the Supplementary Tables of Kar et al, which they give mean number of ori at birth for different media:

0.5% glucose: $\langle T_d \rangle = 113$ min, $\langle n_{ori} \rangle = 1.98$;

0.3% glycerol + trace elements: $\langle T_d \rangle = 148$ min, $\langle n_{ori} \rangle = 1.60$.

While we have no reason to dispute this data, it does not match what we observe for our conditions (0.2% glucose, $\langle T_d \rangle = 133$ min, $\langle n_{ori} \rangle = 1.14$).

Furthermore, over the years many studies on the E. coli chromosome organisation have looked at cells with a doubling time of ~130 min and also describe cells as being born (primarily) with a single

ori: Bates & Kleckner 2005, Cell (125 min), Nielsen et al 2006 Mol.Micro 61 (115 min), Kuwada et al 2013 NAR (120 min), Mäkelä et al PNAS 2021 (~150 min). On the other hand at a doubling time of ~60 min most cells are born with 2 ori (Youngren et al 2014 Genes Dev, Walden et al 2016, Cell).

We put this disparity down to differences in the growth conditions (the concentrations of glucose and glycerol used in the cited works) and strains used (BW27783 of Kar et al and Tiruvadi-Krishanan et al compared to the more common strains used for chromosome studies MG1655 and AB1157).

In any case, we have now removed the phrase “and surprisingly given the slow growth rate”.

4) “It was recently suggested that *ter* centralisation shortly precedes the stable appearance of a constricted/bilobed nucleoid structure (Männik et al., 2016)” – This is misquotation of Männik et al., 2016, which states “The permanent constrictions, which were present uninterrupted until cells divided, appeared in all strains only after the *Ter* region centralized (Figure 5C).” The transient nucleoid constrictions were seen before *Ter* centralization. Please correct the whole paragraph.

We do not understand the point here. We refer to the *stable*, i.e. permanent, appearance of nucleoid constriction. We find that it “occurs significantly later (45 min) than *ter* centralisation”. In Männik et al, continuous/permanent (both terms are used) constriction occurs only about $0.07 \cdot T_d = 8$ min after *Ter* centralisation (Fig. 5C). This is a significant difference.

We did not quote Männik et al. We simply described in our own words the 8 min difference of Männik et al as “*ter* centralisation shortly precedes the stable appearance of a constricted/bilobed nucleoid structure”.

In any case, a very recent preprint (Govers et al, 2023 biorxiv, 10.1101/2023.01.16.524295) has shown that growth media has a substantial effect on when in the cell cycle the nucleoid constricts or indeed initiation occurs through the change in cell size. Therefore the difference between our results and that of Männik et al. is likely due to the different growth conditions used (M9+0.3% glycerol @ 28C vs M9+0.2% glucose @ 30C). The paragraph has been rephrased to reflect this.

5) “Overall, these results confirm that the migration of *ter* to mid-cell, and its maintenance there, does not require the linkage of the terminus region to the divisome” – This was also shown in Mannik et al 2016.

Indeed and in Espeili et al 2012. We acknowledge these papers and use the word ‘confirm’. We now make this more explicit by writing “Overall, these data confirm previous results that”.

6) Section “Quantitative timings of cell cycle events” – discussion why timings have been studied instead of cell lengths is more appropriate to the Results or Materials and Method section.

We have moved the paragraph to the methods as requested.

7) “Surprisingly, given the mean 133 min cell cycle duration, we found that 15% of cells were born with two ori foci, the majority of which do not initiate replication within their cell cycle (Fig. S1F).” –

The argument that the mean doubling time is 133 min in this growth condition is somewhat concerning. It might imply phototoxic effects. Some control with unlabeled strain is needed.

Since Tiruvadi-Krishanan et al., cited above, measure a cycle duration of 113 min using 0.5% glucose at 28C, we do not think it is unexpected that under our conditions of 0.2% glucose at 30C, we achieve a duration of ~133 min.

Govers et al 2023 use the same media and indeed find two ori at birth (as assessed by SeqA foci). However, they grow cells in liquid culture at 37C and achieve a growth rate of 0.75 per hour (55 min doubling). We grow at 30C in a mother machine device and obtain a grow rate of 0.31 per hour (133 min) while imaging.

While it is true that phototropic effects (from blue light to visualise mTurquoise2-ParB) occur, as we stated in the methods, the only effect we observed was a relatively minor 13% increase in cycle duration and all strains imaged were subject to the same blue-light exposure even when not required.

In this growth condition, the majority of cells should have two oris at birth.

As we stated above, we do not observe this and we see no inconsistencies in our data sets, which are large (tens of thousands of cell cycles) and obtained from steady-state conditions.

It is expected that because of noise in the initiation timing, a fraction of these cells “do not initiate replication within their cell cycle”.

We agree that the timing of initiation is subject to noise. However, it is not quite true that a fraction of cells do not initiate in their cell cycle, at least not a random fraction. If this were true anucleate daughters would be produced. Rather, only cells born with two or more ori do not initiate within their cell cycle and we discussed this in more detail in a recent work (Köhler et al, 2023, Biophysical Journal). A similar observation was made for faster growth conditions in Fig. S2 of Si et al 2019 (Curr. Bio 10.1016/j.cub.2019.04.062) and the authors felt this was surprising enough to state “This result disputes the hypothesis that a cell always ensures one-to-one initiation-division correspondence for every division cycle.”

I would have expected some conclusion about the ori-ter dynamics rather than going tangentially to a discussion of double initiations.

This description is inaccurate. We did discuss ori-ter dynamics. Furthermore double initiations were only discussed in the last paragraph of the last subsection of the discussion. However, we have now improved the first discussion to make clearer that we believe our results are most consistent with entropy-driven segregation.

Reviewer #3 (Remarks to the Author):

Overall impression

Sadhir and Murray present an interesting paper addressing the age-old question of how bacterial chromosomes segregate. Experiments are focused primarily on understanding segregation of the terminus region. This is understandable, as *ter* exhibits dramatic movements and is the only chromosomal locus (in *E. coli*) known to attach to cell structure capable of directing movement. Positions of *ori*, *ter*, and occasionally the nucleoid, are measured in slow growing cells by time lapse imaging in a microfluidics slide. Although there have been (many!) studies of *ori* and *ter* dynamics in twenty plus years, their approach is unique in that individual cells are examined over time with high temporal resolution. In addition, they avoid two pitfalls that have plagued many studies: dimeric fluorescent tags and unusual strain backgrounds. Analyses are sophisticated and adhere to a terminus-specific objective throughout the manuscript. They convincingly show that positioning of *ter* to midcell, an event presumed to be prerequisite for partitioning of daughter chromosomes at division, is independent of MatP-mediated linkage to the division apparatus but dependent on segregation of the origin region. Both of these findings are unexpected and important. Claims that origin segregation is only required for *ter* positioning (not the bulk nucleoid), and that chromosomes are oriented longitudinally and not transversely, are less convincing. Writing is crisp, and the authors have an uncanny knack for addressing questions and counter points at the moment they occur to the reader.

Major comments

1. Claims that delaying *oriC* segregation (using a dimeric CFP tag) does not disrupt bulk nucleoid segregation are unsupported by the data. While the effects on *ter* centralization are clear, there is no examination of nucleoid segregation per se. This is a major point in the paper, addressed in the results and discussion. Segregating nucleoids are even drawn in Figure 6B (although labelled “*ori*”), as if nucleoids were examined. HU or DAPI imaging should be performed to support this claim. Alternatively, other intermediate loci could be labelled, although this would be unnecessarily difficult unless the tools are already in hand. If possible, showing this data in supplemental would be useful to the field.

Our thinking here was that successful chromosome segregation is shown by the fact that both daughter cells grow and divide normally and therefore must have inherited a (complete) chromosome from the mother cell (as well as the fact that we observe *ori* and *ter* foci in the daughters). We had shown the distribution of growth rates and cycle durations in Fig. S8. We now show example images of the growing daughters (Fig. S9).

However, to assess nucleoid segregation more directly and confirm that impaired *ori* segregation does not disrupt nucleoid segregation, we can turn to our triple-labelled WT strain (that has nucleoid labelling). In this strain, we still see that ~0.7% of cells have an inverted orientation. The data set is so large (33593 cell cycles) that this fraction constitutes over a hundred cells. We now provide some random examples of these subset of cells and their daughters (new Fig. S9).

2. The longitudinal versus transverse debate (if there is one) is not answered by this paper. Like the early FISH studies proposing the longitudinal model with circular chromosome oriented sideways in the cell “*ori*-arms-*ter*”, data in the current paper are restricted to *ori* and *ter*. The transverse model (“left arm-*ori*-right arm”) on the other hand, was born from imaging several arm loci. The fact is, most data examining interstitial loci support the transverse data. I strongly advise softening anti-transverse-model statements in the discussion section “the organization of the

chromosome”, and instead focusing on how the current data confirms and extends the early ori/ter work and how it fits into the greater evolution of our view of E. coli chromosome organization.

Firstly, we would like to point out that the transverse studies do not simultaneously image ori and arm locus but rather two opposite arm loci and they do not image the nucleoid. Therefore, only the position of the loci within the cells are measured. As we and others have shown the nucleoid is not always centrally positioned within the cell. Indeed, it is not unexpected that it is closer to the new pole at birth. So these studies also have their limitations. Secondly, two high-throughput quantitative studies (Youngren et al 2014, Cass et al 2016) of faster growth rates (~1 hour) that label arm loci (albeit singly) support a longitudinal organisation. (These should have been mentioned in the corresponding results section and not only in the discussion.)

All that said, we have tried to be diplomatic in our statements and simply describe our results. We presume that this reviewer is referring to our statement “Indeed, our work shows that a longitudinal organization can occur during slow growth.” but we immediately qualified that statement by clarifying that “ori are not located at the extreme edge of the nucleoid but close to the outer quarter positions i.e. 25% of the nucleoid mass lies between each ori and the closest pole.” However, we agree that more evidence is required to support a longitudinal organisation. Therefore, at the risk of starting further debate, we have examined strains in which the ori, the nucleoid and a left or right locus are labelled (as well as both left and right without ori). The results (new figure S3) support our claim that **under our conditions**, the chromosome is longitudinally organised.

To be clear, we do not believe that the previous transverse studies are wrong, our results simply indicate that a longitudinal-like organisation is also possible at slow growth. Since, we see similar results using different media (same as used in one of the previous transverse studies), we can only speculate that our observations are due to the nature of growth inside the mother machine, which may be more similar to growth in liquid culture than to growth on agarose pads, and conversely the growth on agarose pads is almost certainly slower than in the corresponding liquid media (on which the given doubling time is usually based.). We have modified the discussion to reflect these points.

Minor comments

1. Results 1st paragraph, please note that monomeric turquoise tagged ParB fusion was chosen to avoid artifacts of dimeric tags. It is widely known that the ParABS system was problematic, but not that it was due to tag multimerization. It should be mentioned here.

Done.

2. It is said that “thousands” of cells are imaged in experiments, but I cannot find n for most figures. Please add this information.

The number of cell cycles are specified in the legend of every figure or by reference to a previous figure e.g. for Figure 1 , we write “Data is from 38066 cell cycles.” The one exception (Figure S5) has been corrected.

3. It is mentioned that variability was lower when relative cell ages are used instead of minutes after the previous division. If so, why isn't relative cell age used throughout? Example, Figures 3 and 4.

We did not actually state that. We wrote in the discussion “However, comparing kymographs (based on relative cell age) and demographs (based on cell length) we saw no indication that cell size is a better metric for studying *ter* centralisation.”

Regarding actual cell age (time) vs relative cell age, for generating the average cell cycle kymographs e.g. Fig. 1B,D we use relative cell age in order to be able to do the averaging. These are meant for visualisation purposes. Relative cell age was not required in Figures 3 and 4 because there we are not interested in the entire cell cycle but only the behaviour around the time point of *ter* centralisation. We do not expect that events like this would occur more rapidly/slowly in cells with shorter/longer cycle durations. Therefore absolute time seemed to us to be the appropriate measurement unit.

4. In the text relating to Figure 2, it is said that old/new pole bias of *ori* position cannot be determined from snap-shot analyses, however, this is not true (in fact it was in the early FISH papers). Polarity can easily be determined by position of *ter*, which is strongly new pole biased.

We do not exactly state that. Yes, if the *ter* is labelled then it can act as a good predictor of polarity but only early in the cell cycle and not after it centralises. In our case, we know the polarity of each cell with certainty. The offending sentences “Note that the bias at birth is only apparent when cells are ordered according to their polarity. It is not detectable when cells are oriented randomly, as would be the case for a snapshot-based analysis” has been changed by the replacement ‘would’ -> ‘may’ to make clear the comparison is of averaging with and without pole information, rather than stating something definitive about snapshot analyses.

5. Figure 3A line colors are not explained.
Corrected.

6. First line in the results section “*ter* centralization coincides with completion of *ori* segregation” cites Figures 1C and S1G, but these are incorrect.
Corrected.

7. 2nd paragraph under “*ter-ori* coupling does not depend on the *ter* linkage. To my knowledge the observation of *ter* overshooting the Z-ring in *matPΔC20* strain is new and very interesting. Elaboration would be good here.

This result was shown previously in Männik et al. 2016. However, in our study we have (more than) enough cells ($n=18202$ compared to $n=18$) to see the average picture clearly. While overshooting does happen, it is not enough to disrupt the overall WT centralisation pattern (notwithstanding the earlier splitting of duplicated *ter* at the end of the cell cycle).

8. Same paragraph, “Fig 6 E,F,H,I” should be designated “S” supplemental.
Corrected.

9. Following paragraph, “consistent with flow cytometry” may not be understandable to many. Please add a brief explanation in the text.
Replaced with “consistent with measurements of DNA content”.

10. Figure 6A, please add percentages for normal, inverted, and mixed classes (as in Fig S7C).

Done but note that the examples cells in Figure 6A all start with the normal orientation so correspond to only the blue coloured populations in the new Fig. S8C.

11. There is a large white space in the Methods section after “ori duplication”. Is there missing text?

This was a formatting error during preparation. Corrected.

12. The distributions of cell cycle durations and growth rates seem wide, but it is difficult to compare to past studies. Comment? Ideally, you would want to compare yours to a study using batch cultures, not microfluidics.

We do not know how single cell cycle durations and growth rates could be measured in batch cultures. In a different microfluidic setup (using a large chamber not microchannels) Wallden et al grew cells at a similar growth rate (mean growth rate 0.0043 per min compared to our 0.0055 per min). They found a similar distribution in growth rates with the same coefficient of variation (CV) of 0.23 (Fig. 2E of Wallden et al 2016, Cell).

Note that cell size studies typically use faster growth rates than we have used and this, along with the associated difference in media and growth conditions, could affect the degree of heterogeneity and indeed the CV appears to be lower at at faster growth rates (Wallden et al 2016, Taheri-Araghi et al 2015, Curr. Bio.).

Reviewer #1 (Remarks to the Author):

The authors responded effectively to reviewers comments. High quality and interesting work. I only have a few minor comments.

Specific remarks

1) Thanks for the line and page numbers.

2) As far as FtsK is concerned; most published evidence for its late action in the septal pore applies to its involvement in the Xer recombination step at dif. Whether or not Z-ring associated FtsK can load/unload and translocate (parts of) the chromosome before cell constriction starts is less clear (to this reader at least). The results with the FtsK ATPase mutant by Galli et al does indicate that FtsK translocation is not needed for Ter centralization. However, the Ter-linkage and FtsK could play redundant roles in centralizing and/or in maintaining Ter centralized, and it would be interesting to know if introduction of the FtsK ATPase (or C-terminus) mutant had any effect on Ter dynamics in delta-zapB or delta-matP cells, for example.

My thinking may be wrong and, given your modifications of the relevant text, I am not demanding this be done for this paper. But, you do have an excellent set-up to test this and it would be nice to definitively rule out (or in?) a role for FtsK in the Ter centralization step, at some point.

3) Lines 284-285. You speculate that 'the final stage of ori segregation somehow triggers ter to rapidly move to midcell'. Of course, the converse possibility is that ter-centralization somehow forces/signals the ori's to not segregate any (or much) further. Can you explain to the readers why you prefer the first possibility?

4) Line 308. You mean Fig. S7B, F.

5) Line 918. Colocalization within 260 pixels? You mean nm (i.e. 4 pixels)?

6) Line 452. Incomplete sentence; ori's do what 'into the cycle of the daughter cells.'?

7) Line 494. Cut 'However'?

8) Line 559. Cut 'Hamamatsu Photonics camera' (redundant).

Reviewer #2 (Remarks to the Author):

Sadhir et al. have mostly responded to my comments. Adding some mechanistic interpretation and modeling would have significantly strengthened the manuscript, though.

The response from the authors to my earlier comment that "We did not quote Männik et al. We simply described in our own words the 8 min difference of Männik et al. as "ter centralisation shortly precedes the stable appearance of a constricted/bilobed nucleoid structure"" is not correct. The text of the manuscript (lines 40-42, page 7) includes an explicit citation to Männik et al., 2016 and does not mention 8 min. The authors should be explicitly state that Männik et al. found stable constriction occurred after the Ter centralization (0.06Td (Td-doubling time) based on Fig. 5C in this paper) and they found that stable constriction occurred 45 min after the Ter centralization. Both works show that the stable constriction occurs after the Ter centralization. The author can, of course, state "In contrast" but the comparison should be accurate and not imply that the order of the events they observe is different than in the cited paper.

Also, the authors should explain how the timing for stable constriction in nucleoid depends upon the threshold they set. How would it have changed if a higher threshold had been chosen? Why the specific threshold value " ...(0.13) is given by the 95th percentile of the relative depth of the nucleoid signal in new-born cells i.e. the first bin of the plot in Figure S3B" was chosen rather than

some other criterion? It is very likely that the difference in threshold explains the quantitative difference between the two works more than the growth rate difference, which is invoked now.

Reviewer #3 (Remarks to the Author):

Major point 1, Lack of nucleoid data. Showing DAPI staining (new Fig S9) satisfactorily indicates that chromosome segregation occurred more or less normally in the examples shown. I could not find in the text a reporting of the frequency of normal DAPI segregation. This should be added if absent.

Major point 2, Weak longitudinal model.

I agree with the authors that the new data showing left and right arm tags supports the longitudinal model. However, I am curious why they chose not to disclose relative positions of the left and right arm tags, which is provided by their dual labeling approach. As clearly shown in their model (Fig S3L), the relationship between L and R tags is the best test of transverse and longitudinal configurations. A color overlay of S3H (left tag) and S3I (right tag) would do this, with accompanying percentages of observed configurations similar to that shown in Fig S3J/K. Additionally, the figure legend to S3 is somewhat confusing, as some strains contain multiple tags, but individual plots show only one tag. Perhaps the legend and figure labeling could be clarified.

Minor points are adequately addressed.

Reviewer #1 (Remarks to the Author):

The authors responded effectively to reviewers comments. High quality and interesting work. I only have a few minor comments.

Specific remarks

1) Thanks for the line and page numbers.

2) As far as FtsK is concerned; most published evidence for its late action in the septal pore applies to its involvement in the Xer recombination step at dif. Whether or not Z-ring associated FtsK can load/unload and translocate (parts of) the chromosome before cell constriction starts is less clear (to this reader at least). The results with the FtsK ATPase mutant by Galli et al does indicate that FtsK translocation is not needed for Ter centralization. However, the Ter-linkage and FtsK could play redundant roles in centralizing and/or in maintaining Ter centralized, and it would be interesting to know if introduction of the FtsK ATPase (or C-terminus) mutant had any effect on Ter dynamics in delta-zapB or delta-matP cells, for example.

My thinking may be wrong and, given your modifications of the relevant text, I am not demanding this be done for this paper. But, you do have an excellent set-up to test this and it would be nice to definitively rule out (or in?) a role for FtsK in the Ter centralization step, at some point.

Thank you for these comments. It is true that FtsK could have an effect on Ter centralisation/maintenance independent of its ATPase activity and we will certainly consider this in our future experiments.

3) Lines 284-285. You speculate that 'the final stage of ori segregation somehow triggers ter to rapidly move to midcell'. Of course, the converse possibility is that ter-centralization somehow forces/signals the ori's to not segregate any (or much) further. Can you explain to the readers why you prefer the first possibility?

This is indeed an alternative possibility. We now explain this and in the discussion we add a sentence explaining that under the hypothesis of entropically-driven rearrangement both schemes are really one and the same thing.

4) Line 308. You mean Fig. S7B, F.
Corrected.

5) Line 918. Colocalization within 260 pixels? You mean nm (i.e. 4 pixels)?
Corrected.

6) Line 452. Incomplete sentence; ori's do what 'into the cycle of the daughter cells.'?
The sentence is not incomplete but could have been better phrased. Corrected.

7) Line 494. Cut 'However'?
Done.

8) Line 559. Cut 'Hamamatsu Photonics camera' (redundant).

Done.

Reviewer #2 (Remarks to the Author):

Sadhir et al. have mostly responded to my comments. Adding some mechanistic interpretation and modeling would have significantly strengthened the manuscript, though.

The response from the authors to my earlier comment that “We did not quote Männik et al. We simply described in our own words the 8 min difference of Männik et al. as “ter centralisation shortly precedes the stable appearance of a constricted/bilobed nucleoid structure” is not correct. The text of the manuscript (lines 40-42, page 7) includes an explicit citation to Männik et al., 2016 and does not mention 8 min. The authors should be explicitly state that Männik et al. found stable constriction occurred after the Ter centralization (0.06Td (Td-doubling time) based on Fig. 5C in this paper) and they found that stable constriction occurred 45 min after the Ter centralization. Both works show that the stable constriction occurs after the Ter centralization. The author can, of course, state “In contrast” but the comparison should be accurate and not imply that the order of the events they observe is different than in the cited paper.

We do not imply that the order is different. Could this reviewer be misunderstanding the meaning of the word ‘precedes’ and the distinction between ‘quote’ and ‘cite’? In any case, we have rephrased the text along the lines the reviewer suggests.

Also, the authors should explain how the timing for stable constriction in nucleoid depends upon the threshold they set. How would it have changed if a higher threshold had been chosen? Why the specific threshold value “...(0.13) is given by the 95th percentile of the relative depth of the nucleoid signal in new-born cells i.e. the first bin of the plot in Figure S3B” was chosen rather than some other criterion? It is very likely that the difference in threshold explains the quantitative difference between the two works more than the growth rate difference, which is invoked now.

If we have no minimum threshold (so that any dip in the middle third of the HU profile would count as a constriction), then constriction of the nucleoid would begin ~25 min before Ter centralisation and often even from birth. However, since we do not smoothen the nucleoid profile, this is the result of noise in the signal and is not at all consistent with visual inspection of the images (noise may be more relevant for us than Männik et al 2016 due to the smaller pixel size of our camera (65 nm vs 160 nm)). For example, the cell used in Figure 3A would be marked as constricted from the 5th frame rather than the 19th, while inspection and thresholding of the images shows this is not the case even for the 17th frame (top most cell shown). That is why we use the 95th percentile of cells at birth. Under our conditions, we see no evidence of nucleoid constriction in newborn cells and so we use the ‘constriction’ depth in these cells as a measure of the noise of the profile. We have now expanded the methods section to make this clear.

Reviewer #3 (Remarks to the Author):

Major point 1, Lack of nucleoid data. Showing DAPI staining (new Fig S9) satisfactorily indicates that chromosome segregation occurred more or less normally in the examples shown. I could not find in the text a reporting of the frequency of normal DAPI segregation. This should be added if absent.

We presume this reviewer is asking for the frequency at which anucleate cells are produced. We see no significant production of anucleate cells. For the triple labelled strain, we found that ~0.5% of cells born with an inverted organisation produced anucleate daughters. This was comparable to the value (~0.25%) for cells born with a normal organisation. We have now added this to the legend of Fig. S9.

Major point 2, Weak longitudinal model.

I agree with the authors that the new data showing left and right arm tags supports the longitudinal model. However, I am curious why they chose not to disclose relative positions of the left and right arm tags, which is provided by their dual labeling approach. As clearly shown in their model (Fig S3L), the relationship between L and R tags is the best test of transverse and longitudinal configurations. A color overlay of S3H (left tag) and S3I (right tag) would do this, with accompanying percentages of observed configurations similar to that shown in Fig S3J/K. Additionally, the figure legend to S3 is somewhat confusing, as some strains contain multiple tags, but individual plots show only one tag. Perhaps the legend and figure labeling could be clarified.

We do not agree that the relationship between L and R tags is the best test of transverse and longitudinal configurations. Given that that nucleoid is not centrally positioned within the cell, one cannot examine, for example, the frequency of L and R being located in opposite cell halves/quarters as that assumes regularly and symmetrically positioned nucleoid lobes. By comparing to the ori and old/new pole as we have done, we can assess the two models using only the ordering of the loci within the cell. What we can do is measure the distance between the loci in the different strains and then compare between them. As shown in the cartoon in S3L, in the longitudinal scheme the loci should be closer to each other than to ori. In the transverse scheme, they should be closer to ori than to each other. We have now added this analysis as well as the requested overlay of S3H and S3I.

We have also clarified the labelling.

Minor points are adequately addressed.